# 𝕄ARBLE: A HARD BENCHMARK FOR MULTIMODAL SPATIAL REASONING AND PLANNING

## ABSTRACT

The ability to process information from multiple modalities and to reason through it step-by-step remains a critical challenge in advancing artificial intelligence. However, existing reasoning benchmarks focus on text-only reasoning, or employ multimodal questions that can be answered by directly retrieving information from a non-text modality. Thus, complex reasoning remains poorly understood in multimodal domains. Here, we present 𝕄ARBLE, a challenging multimodal reasoning benchmark that is designed to scrutinize multimodal language models (MLLMs) in their ability to carefully reason step-by-step through complex multimodal problems and environments. 𝕄ARBLE is composed of three highly challenging tasks, 𝕄-PORTAL, 𝕄-CUBE and 𝕄-MAZE, that require the crafting and understanding of multistep plans under spatial, visual, and physical constraints. We find that current MLLMs perform poorly on 𝕄ARBLE—all 12 advanced models obtain around 0% accuracy performance on 𝕄-CUBE and 𝕄-MAZE, while only Grok-4 and GPT-5 slightly outperformed the random baseline on 𝕄-PORTAL. These results indicate that complex reasoning is still a challenge for existing MLLMs. Moreover, we show that perception remains a critical bottleneck to multimodal reasoning. By shedding light on the limitations of MLLMs, we hope that 𝕄ARBLE will spur the development of the next generation of models with the ability to reason and plan across many multimodal reasoning steps.

## 1 INTRODUCTION

Human reasoning is inherently multimodal and sequential—integrating modalities such as language or vision as context to draw conclusions through structured, step-by-step thought. While LLMs have made significant strides in step-by-step reasoning (Wei et al., 2022; Jaech et al., 2024; Guo et al., 2025; OpenAI, 2025), the multimodal reasoning abilities of Multimodal LLMs (MLLMs) are still in their infancy and not yet well understood. Achieving complex, multi-step, multimodally grounded reasoning is critical for building intelligent systems that can generalize across domains and interact adaptively with complex environments.

Recent benchmarks – such as ScienceQA (Lu et al., 2022), MathVista (Lu et al., 2023b), and MMMU (Yue et al., 2024) – have shown that MLLMs can solve tasks involving both visual and linguistic understanding. However, these benchmarks often emphasize relatively shallow forms of reasoning, such as single-step question answering or factual retrieval. They frequently conflate *perception* (*e.g.*, interpreting an image or diagram) with *reasoning* (*e.g.*, drawing logical inferences, comparing evidence, or crafting a multi-step plan), reducing complex reasoning to pattern matching and multimodal integration. As a result, current evaluations underexplore and undermeasure an MLLM's capacity for deep, structured reasoning. Moreover, the recent literature has focused heavily on abstract reasoning in domains such as advanced mathematics or code generation, where multimodal embodiment plays a limited role. In contrast, interacting with and planning in spatially and physically constrained environments is a fundamental dimension of human intelligence but it is largely missing from today's MLLM evaluations. While a recent effort introduced an escape room-inspired benchmark (Wang et al., 2025b), frontier models were not sufficiently challenged by its task complexity, achieving up to 100% escape rate. Thus, hard benchmarks that stress multi-step planning and spatial reasoning under physical constraints remain an open need. Analogous to how difficult challenges have historically driven progress, we believe that an ARC-like test (Chollet et al., 2024) for multimodal reasoning could spark foundational advances in MLLM capabilities.

Table 1: Conceptual overview of the $\mathbb{M}$ARBLE benchmark.

| Dataset | Description | Subtasks | # Samples | Metrics |
|---------|-------------|----------|-----------|---------|
| $\mathbb{M}$-PORTAL | Solving complex multi-modal spatial reasoning and planning problems. | `Plan correctness,` `Fill-the-blanks` | 512 512 | F1-Score, Accuracy |
| $\mathbb{M}$-CUBE | Assembling 3D Cube from six jigsaw pieces. | `CUBE,` `CUBE-easy` | 1,000 1,000 | Accuracy |
| $\mathbb{M}$-MAZE | Solving dynamic mazes by combining tile insertion and player navigation. | `MAZE,` `MAZE-easy` | 1,000 1,000 | Success Rate |

In this work, we present $\mathbb{M}$ARBLE (MultimodAl Reasoning Benchmark for Language modEls), a highly challenging multimodal reasoning benchmark specifically designed to evaluate step-by-step, multimodally grounded reasoning in MLLMs. Our benchmark introduces tasks that are cognitively demanding, requiring models to decompose complex multimodal prompts into interpretable intermediate steps, align information across inputs, and to carefully craft a multi-step plan to solve complex problems under diverse spatial and physical constraints. Unlike prior datasets that overemphasize final-answer accuracy, our benchmark emphasizes reasoning trajectories and plans, providing both gold-standard rationales and mechanisms for evaluating intermediate step fidelity. $\mathbb{M}$ARBLE consists of three main tasks, $\mathbb{M}$-PORTAL which tests complex spatial reasoning and planning abilities, $\mathbb{M}$-CUBE, which tests the ability to understand and assemble 3D jigsaw pieces into a target cube shape, and $\mathbb{M}$-MAZE, which test the ability to plan the path to target in an editable maze. Each dataset also contains two subtasks at different difficulty levels, as shown in Table 1.

We conduct an extensive evaluation of $\mathbb{M}$ARBLE across 12 state-of-the-art MLLMs and reasoning models. Intriguingly, most models obtain near-random performance on $\mathbb{M}$-PORTAL and around $0\%$ accuracy on $\mathbb{M}$-CUBE and $\mathbb{M}$-MAZE. Even in simplified configurations, only about half of the models are able to outperform the random baseline. Notably, Grok-4 and GPT-5 are the only model demonstrating reasonable performance on $\mathbb{M}$-PORTAL, achieving $18.2\%$ and $14.2\%$ F1 score, respectively. However, they still completely fail on the harder tasks of $\mathbb{M}$-CUBE and $\mathbb{M}$-MAZE. These results indicate that complex multimodal reasoning remains a significant challenge for current MLLMs. Our further analysis shows that perception is still a bottleneck for multimodal reasoning: all the advanced MLLMs completely fail to understand and extract structured information from the visual inputs. Additionally, we present an interactive setup for $\mathbb{M}$-CUBE and $\mathbb{M}$-MAZE to help the multimodal reasoning via the feedbacks from the environments, reflecting the real-world and agentic problem-solving processes. We hope that $\mathbb{M}$ARBLE will serve as a probing benchmark to reveal the limitations of current MLLMs and drive the development of next-generation models with stronger capabilities in multi-step multimodal reasoning and planning.

## 2 $\mathbb{M}$ARBLE: A BENCHMARK FOR MULTIMODAL SPATIAL REASONING AND PLANNING

We present $\mathbb{M}$ARBLE, a challenging game-inspired multimodal reasoning benchmark designed to evaluate the complex reasoning abilities of multimodal LLMs (MLLMs). In contrast to prior reasoning benchmarks that evaluate only the final answer independent of the reasoning trace, $\mathbb{M}$ARBLE focuses on assessing the correctness of the reasoning process itself. $\mathbb{M}$ARBLE consists of three tasks, $\mathbb{M}$-PORTAL, $\mathbb{M}$-CUBE and $\mathbb{M}$-MAZE, all require complex, multi-step and multimodal reasoning skills to forge an appropriate plan that accounts for complex spatial and physical problem constraints. The $\mathbb{M}$-PORTAL task challenges MLLMs to solve problems derived from Portal 2 videogame with multi-step reasoning and planning. The $\mathbb{M}$-CUBE evaluates MLLMs in their ability to solve Happy Cube puzzles, *i.e.*, rotate complex shapes to arrange them into 3D cubes under physical constraints. Finally, the $\mathbb{M}$-MAZE tests the ability of MLLM to plan the correct path to the target, in a dynamic and editable maze.

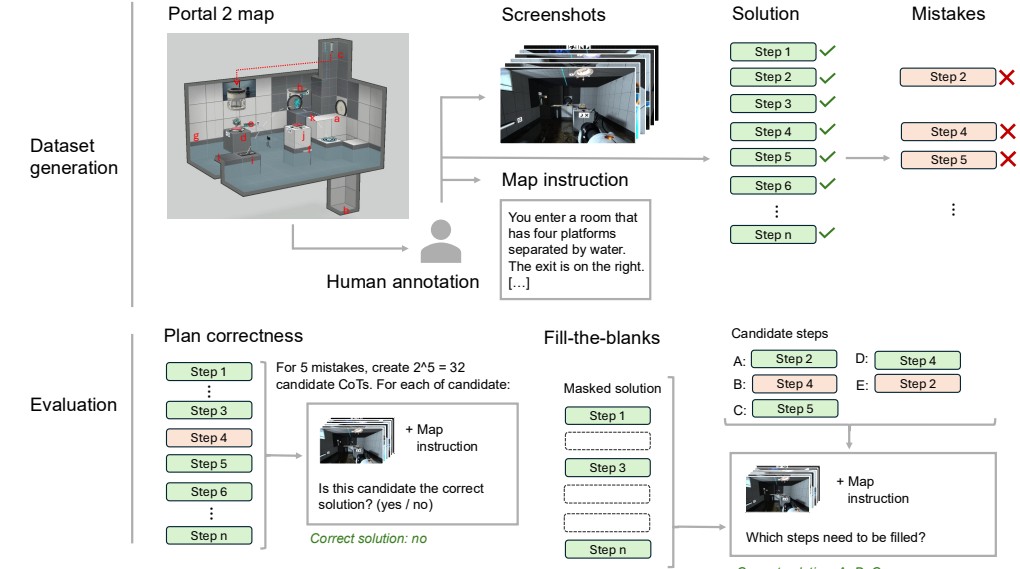

Figure 1: Data generation and evaluation pipeline for the $\mathbb{M}$-PORTAL task. The top row illustrates how a given Portal 2 map (sourced from the community test chambers) was analyzed with human annotation to produce a set of illustrative screenshots that fully depict the map, textual map instructions, a ground-truth solution chain of thought (CoT), as well as a set of five mistaken steps. The steps are designed to operate independently so that mistakes and correct steps can be easily combined. The bottom row indicates two evaluation types of $\mathbb{M}$-PORTAL: first, `plan correctness`, a binary evaluation where candidate solutions have to be rated as correct or wrong. Second, a `fill-the-blanks` evaluation, where multiple steps of the ground truth CoT solution are masked, and multiple options are available to fill in at the right place.

## 2.1  ⚡ $\mathbb{M}$-PORTAL

The $\mathbb{M}$-PORTAL task is a multimodal reasoning task that involves planning, spatial reasoning, as well as multimodal integration. $\mathbb{M}$-PORTAL is inspired by the game Portal 2, a first-person perspective puzzle videogame released by Valve in 2011. Portal challenges players to overcome obstacles and to pass through rooms by means of placing two portals through which players can teleport. A key mechanic in Portal is the conservation of momentum: when a player enters one portal with a given velocity, they exit the second portal with the same relative momentum. This enables creative traversal strategies, such as jumping across large gaps or over obstacles, by combining gravity-driven falls with portal placement. Various additional features (*e.g.*, buttons, lasers, tractor beams, liquids) add further complexity to the puzzle environments. The ultimate trial will be for MLLMs to interactively navigate and solve the game. However, to enable broad accessibility and usability of this benchmark, we abstract a given map into a set of visual question-answering tasks that require the MLLM to integrate several depictions of the map, a textual instruction to the map, in order to examine partial or complete chain of thought (CoT) solution plans that may consist of dozens of steps. Figure 8 in Appendix D gives an introductory overview of how a basic portal map could look like, displaying a scene overview (top left), the step-by-step solution, and a few in-game screenshots.

**Problem statement.** Given an input $X = (\mathcal{I}, T)$, where $\mathcal{I}$ is a set of multimodal inputs (*e.g.*, screenshots of a Portal map or textual contextualization of the environment) and $T$ is a task instruction, the objective is to generate a Chain-of-Thought (CoT) plan $P = (s_1, s_2, \ldots, s_n)$ consisting of interpretable, physically sound reasoning steps that, if executed, would successfully solve the problem. The reward of a plan $R(P)$ is 1 if the exit door is passed, and 0 otherwise. Then the objective is to evaluate the ability of models to implement the mapping $F^*$ that maximizes the reward, *i.e.*,

$$F^* = \arg\max_F \mathbb{E}_{X \sim \mathcal{D}} \left[ R(F(X)) \right], \quad \text{where} \tag{1}$$

$$F : X \mapsto P = (s_1, s_2, \ldots, s_n). \tag{2}$$

**Data collection.** For data collection, a human annotator with advanced Portal 2 experience browsed through top-rated maps from the Portal 2 community test chambers. We focused on the community test chambers, as they were often self-contained, well-defined problems in a single room. The annotator selected 16 high-quality maps that received top user-rating, while being compactly shaped such that they would be amenable to capture within a few screenshots. Figure 1 gives an overview of how the $\mathbb{M}$-PORTAL dataset was created in the top row, whereas the bottom row indicates the evaluation strategies employed in the $\mathbb{M}$-PORTAL task.

**Evaluation subtasks.** Since direct execution and success validation in the Portal environment would depend on a closed-source game environment and could involve a brittle interfacing and limited accessibility, we focus on evaluating the ability of a model to reason about the correctness of candidate plans or the missing steps in incomplete plans. For this, we consider two types of closed-ended evaluations: `plan correctness` and `fill-the-blanks` tasks, each contributing to 512 problems.

1. **Plan correctness:** *Is the provided candidate plan correct?*

   Plan correctness is the binary classification task and requires answering yes/no questions. It is a harder task compared to fill-the-blanks because models have to carefully review lengthy candidate plans that may be dozens of steps long and involve various spatial and physical constraints and dependencies. These candidates may contain no mistake at all up until five mistaken steps. This task has a significant class imbalance, as one Portal map with five available mistaken steps allows the creation of $2^5 = 32$ candidates that leverage individual mistakes, whereas only one out of 32 candidates is correct.

2. **Fill-the-blanks:** *Can the model accurately identify several missing steps given surrounding context and a few candidate options?*

   On the easier fill-the-blanks task, models receive a partial plan to solve the Portal map whereas several steps are masked. To fill the missing steps, the model needs to choose five correct options from five mistake or distracting options in a correct order. Even though this task is hard for a naive random baseline, for a model that is able to interpret the multimodal inputs $X$ as well as the partial solution, it should be easier to identify the correct missing steps especially since mistaken steps also appear in their correct version as highly similar options. Furthermore, fill-the-blanks can also be seen as a simplification as it helps the model focus its attention on a few relevant steps out of a large sequence, whereas in the binary evaluation any step could be potentially mistaken.

## 2.2 🧊 $\mathbb{M}$-CUBE

**Problem statement.** The $\mathbb{M}$-CUBE task is a 3D spatial puzzle inspired by the Happy Cube, a mechanical puzzle originally invented by Dirk Laureyssens in 1986. In this task, one is presented with 6 jigsaw-style pieces taken from the faces of a $5 \times 5 \times 5$ Cube. Each piece is featured by the bump and gap pattern on its edges. The goal is to assemble the pieces into a valid cube where the edges are aligned seamlessly without gap or overlap. To solve the $\mathbb{M}$-CUBE task, an MLLM needs to assign each piece into a cube face with proper orientation, *i.e.*, to rotate and/or flip the piece accordingly to align with other pieces. For each problem, an MLLM must account for 6! possible piece-to-face assignments (modulo rotational symmetries), and for each piece, 8 discrete states of rotations and flips, resulting in a combinatorial explosion of candidate solutions. Among the vast search space, only very few solutions are valid given the geometric constrains imposed by the interlocking bump and gap patterns. András et al. (2013) reported that most commercially available cubes have only one solution (up to rotational equivalences), making this a challenging reasoning problem.

**Data generation.** While the $\mathbb{M}$-CUBE tasks are inspired by the Happy Cube puzzle, we generate all samples synthetically. Figure 2 gives an overview of the workflow. Specifically, the data generation pipeline starts with a $5 \times 5 \times 5$ cube and disassembles the surface into 6 interlocking pieces. Each piece can be regarded as a $5 \times 5$ grid, where the center $3 \times 3$ region is always preserved. For remaining cells located on the edges, we randomly assign each cell to one of the adjacent faces of the big $5 \times 5 \times 5$ cube, to create the bump and gap patterns along the boundary. After that, the obtained pieces are shuffled and rendered from a random 3D viewpoint as the input to an MLLM. We interactively selected viewpoint ranges such that the shape was clearly discernible. Concretely, we

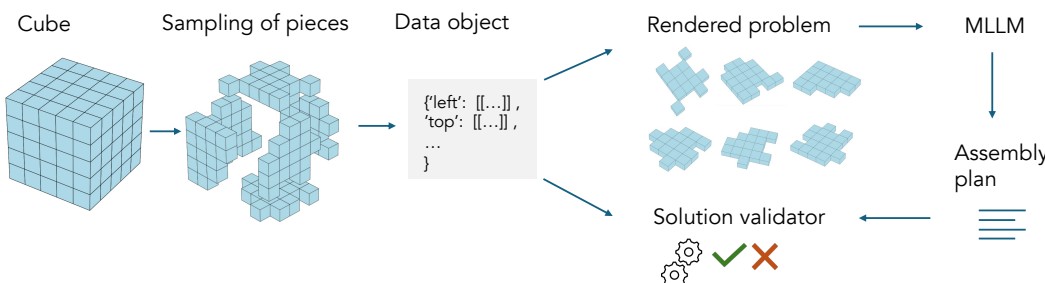

Figure 2: Overview of the $\mathbb{M}$-CUBE workflow including data generation, problem rendering, as well as solution validation.

render the objects by sampling a camera elevation in the range of –155° to –115° and an azimuth in the range of –150° to –90°, relative to the canonical front view. The base view corresponds to an elevation of –135° and an azimuth of –120°, with uniformly random perturbations of ±20° and ±30°, respectively.

**Solution validator.** The model is required to find the correct piece-to-face mapping and the orientation of 6 pieces. However, for each problem, there is no unique solution since a cube contains 24 rotational symmetries. Therefore, instead of directly comparing the answer to ground-truth, we provide a solution validator by testing whether the solution from MLLM could successfully assemble the pieces into a perfect cube. Beside binary evaluation, the solution validator could also identify the conflicts in a given configuration, such as mismatched edges. This diagnostic feedback can be used by an MLLM to iteratively refine its solution. See Appendix D.2 for example.

**Evaluation subtasks.** To measure the performance of MLLMs with controlled difficulty level, we create two subtasks called CUBE and CUBE-easy. Each subtask contains 1000 examples. CUBE-easy is a simplified version of CUBE along three axes: *i)* the input pieces are represented as 2D arrays instead of the rendered image to reduce the perception error of MLLM (see the discussion in Section 3.2 for more details); *ii)* each puzzle is specially designed such that the solution does not require flipping of any pieces; *iii)* a partial solution with the arrangement of 4 pieces is provided in the prompt, leaving only 2 missing pieces to be placed. Consequently, *ii)* and *iii)* significantly reduce the size of search space. In comparison, CUBE retains the full complexity of the task, where the MLLM needs to understand the input images, and explore over all the possible arrangements of the 6 pieces.

## 2.3 🔲 $\mathbb{M}$-MAZE

**Problem statement.** The $\mathbb{M}$-MAZE task is 2D spatial–planning puzzle directly inspired by *The aMAZEing Labyrinth* board game. Each game contains a $7 \times 7$ maze and one off–board *spare* tile. The tile contains three shapes I/L/T and can have different orientations on the board. There are two types of actions in the action space: (i) *insert* the spare tile into one row or column to shift the whole line (ii) *move* along connected corridors. The *insert* action will change the connectivity of the board and make the maze dynamic. Given a board image, a model must produce a valid multi–turn plan to move the player to the target, which poses unique challenges to MLLMs in terms of perception and multi–step reasoning.

**Data generation.** Similar to $\mathbb{M}$-CUBE, we synthesize $\mathbb{M}$-MAZE tasks by generating initial board configurations, starting with 16 fixed path tiles and 12 fixed treasures, then sampling the remaining I/L/T tile shapes, random player positions, and 12 scattered treasures to complete the board. The process begins with board sampling, followed by BFS to compute all trajectories to each target via TILE INSERTION (shifting rows/columns) and PLAYER MOVE (along connected tiles), determining minimal depth $D$ (the fewest turns to reach a target). We subsample trajectories by $D$, a difficulty proxy since higher $D$ increases the search space and planning complexity, and retain one solution per *(board, seed, depth)* triplet for diversity. Evaluation uses only the initial configuration (board grid, player position, and target, excluding other objects to reduce clutter), providing a lower bound on the planning depth required to solve the puzzle.

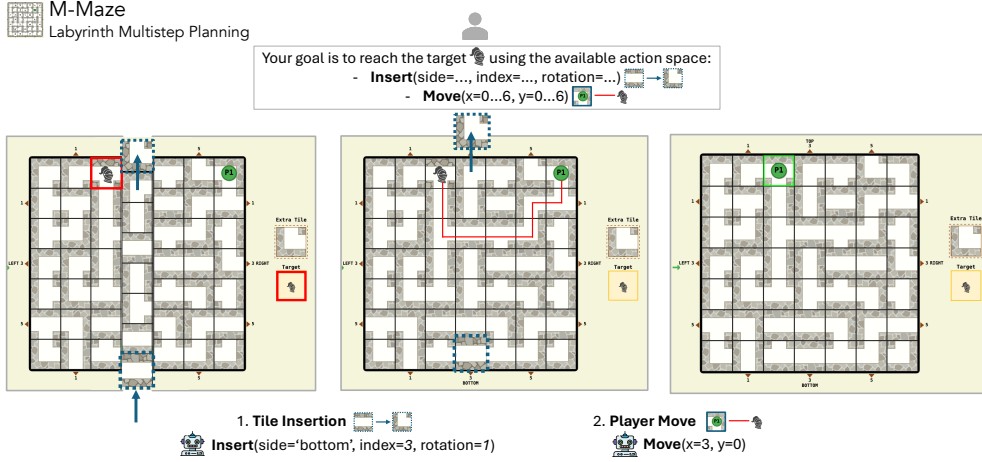

Figure 3: Overview of the $\mathbb{M}$-MAZE task.

**Evaluation subtasks.** Similar to $\mathbb{M}$-CUBE, we create two subtasks to measure the performance of MLLM with controlled difficulty levels: MAZE and MAZE-easy. Each subtask contains 1000 examples. MAZE-easy is a simplified version of MAZE along two axes: *i)* the input board includes a **visual harness** overlay (tile types and coordinates) and the full symbolic state (board grid, object grid, extra tile, player position); *ii)* a reduced depth $D = 2$. These adjustments minimize perception errors and shrink the search space. In contrast, MAZE retains full complexity at depth $D = 4$, requiring the MLLM to interpret the raw board image demanding deep planning and strong visual parsing capabilities.

## 3 EXPERIMENTS

We evaluate performance on the $\mathbb{M}$ARBLE benchmark using eight state-of-the-art MLLMs, including both open-source and closed-source models with advanced multimodal reasoning capabilities. Specifically, we assess three representative open-source MLLMs: Qwen2.5-VL-72B (Bai et al., 2025), InternVL3-78B (Zhu et al., 2025) and Llama-4-Scout (Meta, 2025), alongside eight closed-weight models: GPT-4o (Hurst et al., 2024), GPT-o3, GPT-o4-mini, GPT-5, Claude-3.7-Sonnet (Anthropic, 2025) Gemini-2.5-pro (Google DeepMind, 2025), Seed1.5-VL Team (2025) and Grok-4. In addition, we also include one text-only model DeepSeek-R1-0528 Guo et al. (2025) in the evaluation. We remove or manually convert the input images into textual descriptions to evaluate the models that only takes text inputs. Besides, we provide evaluation of experienced human players on all the tasks. All the experiment configurations, prompts and hyperparameters are detailed in the Appendix E. Experiments are conducted on a single node server with 8 Nvidia H200 GPUs. The overall results are repoted at Table 6.

### 3.1 RESULTS ON ⚑ $\mathbb{M}$-PORTAL

We evaluate state-of-the-art MLLMs on the plan correctness and fill-the-blanks tasks of the $\mathbb{M}$-PORTAL, as reported in Table 6. On the plan correctness task, all the investigated models (except GPT-5 and Grok-4) performed very poorly with a minority class F1 score of around 6%, similar to the random baseline. In comparison, on the easier fill-the-blanks task, 8 out of 12 models outperform the random baseline. In particular, the performance gap compared to the random baseline is substantial ($\geq 20\%$) for Gemini-2.5-pro, GPT-o3, Grok-4 and GPT-5 that significantly outperforms all other models. Interestingly, the best performing model, Grok-4, manages to correctly solve only 46.7% of the problems on fill-the-blanks tasks and achieves 18.2% F1 score on the $plan-correctness$ binary classification. Note that although the fill-the-blanks task results in random baseline scores, it is expected to be easier than the plan correctness task for models capable of interpreting the multimodal inputs and leveraging the partial solution. Also, it's worth noting that the experienced human player could obtain 37.5% on the fill-the-blanks subtask, surpasing all the frontier models except Grok-4.

Table 2: Performance of state-of-the-art MLLMs on the $\mathbb{M}$ARBLE benchmark and three tasks: $\mathbb{M}$-PORTAL, $\mathbb{M}$-CUBE and $\mathbb{M}$-MAZE. Each task contains two difficulty levels. We report F1-score (%) for binary evaluation (plan correctness) of $\mathbb{M}$-PORTAL and success rate (%) for all the other tasks. Human performance was evaluated with 2–3 experienced players on each task. *All the visual inputs are removed or converted to texts for text-only LLMs.

| Models | $\mathbb{M}$-PORTAL | | $\mathbb{M}$-CUBE | | $\mathbb{M}$-MAZE | |
|---|---|---|---|---|---|---|
| | Binary | Blanks | CUBE | CUBE-easy | MAZE | MAZE-easy |
| 👤 *Human* | - | 37.5 | 0.0 | 85.0 | 55.0 | 80.0 |
| 🔀 *Random* | 6.1 | 3e-3 | 1e-5 | 3.1 | 5e-9 | 1e-4 |
| *Open-weights models* | | | | | | |
| Qwen2.5-VL-72B | 6.6 | 0.0 | 0.0 | 2.0 | 0.0 | 0.1 |
| InternVL3-78B | 6.4 | 0.0 | 0.0 | 2.8 | 0.0 | 0.0 |
| Llama-4-Scout | 6.5 | 0.0 | 0.0 | 1.6 | 0.0 | 0.3 |
| Seed1.5-VL | 7.6 | 4.1 | 0.0 | 2.0 | 0.0 | 0.0 |
| DeepSeek-R1-0528* | 0.0 | 10.0 | 0.0 | 8.0 | 0.0 | 2.0 |
| *Closed-weights models* | | | | | | |
| Claude-3.7-Sonnet | 6.3 | 8.8 | 0.0 | 7.4 | 0.0 | 1.0 |
| Gemini-2.5-Pro | 4.7 | 20.0 | 0.0 | 11.0 | 0.0 | 20.0 |
| GPT-4o | 6.5 | 0.4 | 0.0 | 2.0 | 0.0 | 0.0 |
| o4-mini | 0.0 | 5.5 | 0.0 | 16.0 | **1.0** | 23.0 |
| o3 | 6.6 | 23.4 | 0.0 | 72.0 | 0.0 | 69.0 |
| GPT-5 | 14.2 | 29.1 | 0.0 | **84.0** | 0.0 | 66.0 |
| Grok-4 | **18.2** | **46.7** | 0.0 | 38.6 | 0.0 | 47.0 |
| Grok-4 Fast | 15.1 | 31.0 | 0.0 | 53.0 | 0.0 | **75.0** |

**Influence of blanks.** In the fill-the-blanks task on $\mathbb{M}$-PORTAL, each question contains multiple steps in the complete solution, and part of them are masked. To systematically understand the impact of missing information, we construct a series of questions where the model is asked to fill $n$ blanks from $2n$ candidate options. We evaluate the performance of Qwen2.5-VL-72B and the result is shown in Figure 4. Notably, the model obtains around $70\%$ accuracy when only a single blank is present. However, the performance declines rapidly as the number of blanks increases, dropping to less than $1\%$ when $n \geq 4$, which indicates the challenges of the subtask under the conditions of extensive missing information.

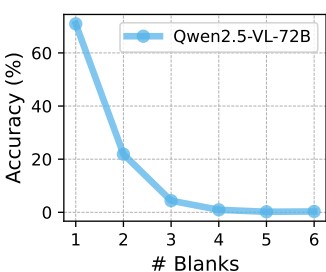

Figure 4: The influence of number of blanks to $\mathbb{M}$-PORTAL.

## 3.2 RESULTS ON $\mathbb{M}$-CUBE

The results on the CUBE and CUBE-easy tasks of $\mathbb{M}$-CUBE are shown in Table 6. Intriguingly, all the advanced MLLMs completely fail on the harder subtask CUBE and obtain $0\%$ accuracy despite more than $10,000$ tokens spent on thinking the problems. The results highlight the complex multimodal reasoning process involved in CUBE, where the model has to iterate over verification and backtracking through a long reasoning chain to make a final answer. In comparison, on the simplified CUBE-easy task, 7 out 12 frontier models are able to perform better than random guess. Among them, GPT-5 and GPT-o3 achieves remarkable performance of $84.0\%$ and $72.0\%$ accuracies, substantially outperforming the remaining models, but are still slightly worse than the human performance of $85.0\%$ accuracy.

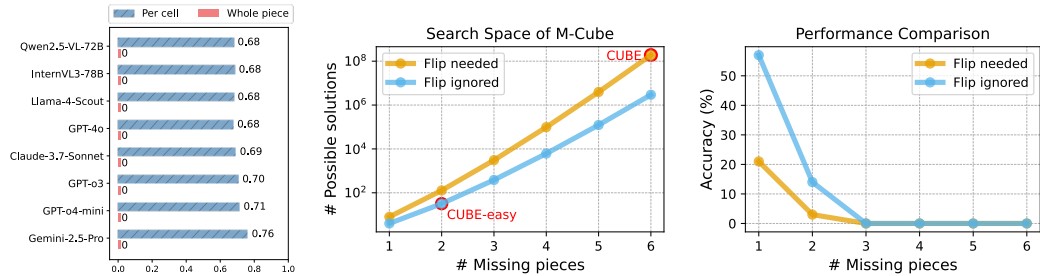

Figure 5: *Left*: Perception remains a bottleneck for 𝕄-CUBE. A perception task designed to test MLLM's ability on retrieve structured information from visual input (full prompt in Appendix D.2). *: Middle*: Search space of the 𝕄-CUBE dataset under different configurations. *Right*: Performance of DeepSeek-R1 across varying levels of task difficulty of the 𝕄-CUBE dataset.

**Error on perception.**    To solve the 𝕄-CUBE puzzle, the first step is to understand the visual input and retrieve the relevant information, which serves as the basis of the reasoning steps afterwards. Thus, we design a perception task to measure whether the MLLMs could correctly extract information from the input image: given a jigsaw-style piece in a 3D viewpoint, the model is asked to convert the piece into a $5 \times 5$ array. We evaluate all the 8 MLLMs on this perception task with 200 test examples, and report the accuracy on cells and accuracy of the whole piece also on Figure 5 left. Surprisingly, we found all the models could only achieve around 70% accuracy per cell. The best perception performance, is 76% accuracy from Gemini-2.5-pro, meaning that the model could still occasionally make mistakes. As a result, all the models achieve 0% accuracy on the whole piece. These results highlight that even advanced MLLMs struggle with this seemingly simple perception task, posing a potential bottleneck for multimodal reasoning in complex scenarios like CUBE.

**Error on reasoning.**    Apart from the perception errors, 𝕄-CUBE still remains a highly challenging problem due to the vast search space from the combination of all possible arrangements and orientations of 6 pieces. Figure 5 illustrates the size of search space of 𝕄-CUBE as a function of both the number of missing pieces and whether a solution requires flipping the pieces. In particular, CUBE comprises $6! * 8^6 = 188,743,680$ possible solutions. In comparison, CUBE−easy only contains 32 possible solutions, a $5,000,000$ fold reduction of the hypothesis space. To isolate the reasoning challenge from perceptual limitation, we manually convert the visual inputs into corresponding text arrays. We then compare the performance of DeepSeek-R1 in different search space configurations, as shown in Figure 5. The model obtains 57% accuracy in the simplest setting with only one missing piece. However, the performance drops drastically as the search space expands, falling to 0% when more than 3 pieces are missing. The substantial decline underscores the difficulty of reasoning among expanding combinatorial search space, a major bottleneck for existing reasoning models. In summary, besides perception error, reasoning among the vast search space is also a challenge, making 𝕄-CUBE an especially difficult task for state-of-the-art MLLMs.

### 3.3    RESULTS ON 𝕄-MAZE

We evaluate state-of-the-art MLLMs on 𝕄-MAZE (MAZE, MAZE−easy) as reported in Table 6. Similarly, all the models performs around 0% on the harder subtask, while on the simper subtask MAZE−easy, GPT-o3, Grok-4, GPT-5 are the models significantly outperforming the other models. Interestingly, there is a clearly performance gap between human player and MLLMs on this task: human achieves remarkably 55.0% on MAZE, 80.0% on MAZE−easy, respectively. Moreover, we observe similar perception bottleneck as 𝕄-CUBE where MLLM struggles on extracting the structured visual information from the input. We defer the empirical results to the Appendix F.5.2.

**Error on Reasoning.**    Beyond perception errors, 𝕄-MAZE challenges models due to the need to reason over state transitions and rules across multiple steps, not just static layouts. To isolate reasoning from perception, we use a *Visual Harness + Symbolic* setup, providing the board state in two forms: a compact symbolic grid as text in the prompt, and the input image with labels overlaid directly onto the board (see Figure 15 in Appendix D). We evaluate GPT-5-MINI, with results in Figure 6: 100% success at $D = 0$, 70% at $D = 1$, 30% at $D = 2$, 15% at $D = 3$, and below

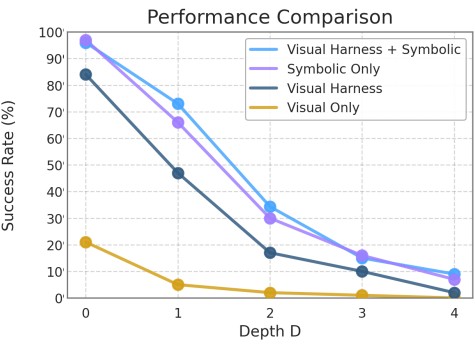

Figure 6: Success rate (%) of GPT-5-mini across depths $D$ on 🎴 𝕄-MAZE, comparing four input settings described in Figure 15 in Appendix D.

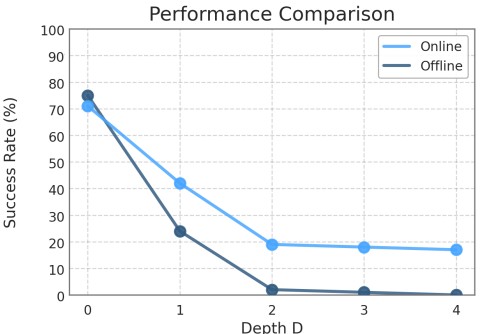

Figure 7: Success rate (%) of GPT-5-nano on 🎴 𝕄-MAZE using *Visual Harness + Symbolic*, across depths, comparing Online vs. Offline settings.

10% at $D = 4$. The steep decline with depth, driven by error accumulation, highlights several of the most frequent failure modes: *(i)* **adjacency misinterpretation errors**, where a model either misjudges non-reciprocal openings as being connected or hallucinates a change in a tile's type to force a valid path, leading to illegal player movement; *(ii)* **state-update errors**, where the model incorrectly processes a row/column shift by failing to also update the positions of players or items on the affected tiles, leading to an incorrect internal representation of the board state; *(iii)* **insert legality errors**, a form of instruction-following error, where models attempt illegal moves like using the wrong slots; and *(iv)* **shallow planning errors**, where a model fails to find a solution and does not output any plan.The steep drop with depth indicates that multi-step reasoning over dynamic, rule-bound states is inherently hard. In summary, beyond perception, planning across multiple turns in a large combinatorial space makes 𝕄-MAZE a challenging task for current MLLMs.

**Online Evaluation**  We evaluate GPT-5-NANO in a per–action loop: at each phase the agent emits one atomic action (INSERT or MOVE); the environment executes it and returns the next observation. Episodes end on success, illegality (no-reverse, lane legality, invalid move), or budget exhaustion. We report **Success Rate@$B$** with $B = 2D$, where $D$ is the optimal depth (two actions per optimal turn: INSERT+MOVE). Results (Fig. 7) show around 80% at $D$=0 and **online surpassing offline once multi-step planning is required**: around 42% vs. 24% at $D$=1, around 19% vs. 2% at $D$=2; online then plateaus at around 17–18% for $D$=3–4 while offline collapses to 0%. Overall, step-wise state updates mitigate error accumulation, but performance still degrades with increasing depth, indicating persistent bottlenecks in multi-step transition modeling, spatial consistency, and rule adherence.

# 4 DISCUSSION

This paper introduces 𝕄ARBLE, a hard multimodal reasoning benchmark for MLLMs. 𝕄ARBLE provides a focused testbed for evaluating MLLMs on complex spatial reasoning and planning tasks that are underlying heterogenous physical constraints. Our tasks are designed such that an MLLM must first understand the physical constraints imposed by the multimodal input, and then formulate a coherent, multi-step plan that draws from a vast search space in order to solve the problem. 𝕄ARBLE fills the gap of multimodal reasoning evaluation by shifting the focus from outcome accuracy to process-oriented, multi-steps reasoning that requires coherent multimodal understanding. By contributing a challenging benchmark for multi-step, multimodal reasoning amidst spatial and physical constraints, 𝕄ARBLE aspires to elicit more progress and innovation in MLLM development that will unlock unprecedented abilities in reasoning and planning amidst complex and multimodal environments—capabilities that are essential for real-world, embodied, and general-purpose intelligence.

Our empirical evaluation reveals that state-of-the-art MLLMs struggle significantly with 𝕄ARBLE. Most of the models can only outperform random baselines in simplified ablations and fail even on structured perception tasks, underscoring limitations in both reasoning and visual understanding.

**Limitations and future work.** We do not explore fine-tuning or adapting models at test time. Future work should investigate adaptive approaches, enabling models to reason *with* and *through* different modalities—such as "thinking with images"—in a more compositional way.

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

## A    LLM USAGE STATEMENT

Generative AI has been used to check for typos and grammatical errors in this manuscript, and to rephrase certain original sentences of the authors for correctness, conciseness and style, as they are not of English mother tongue. Any use of generative AI in this manuscript adheres to ethical guidelines for use and acknowledgment of generative AI in academic research. Each author has made a substantial contribution to the work, which has been thoroughly vetted for accuracy, and assumes responsibility for the integrity of their contributions.

## B    ETHICAL STATEMENT

As with any benchmark, there is a risk of overfitting to dataset-specific patterns. However, our setting involves abstract puzzle domains, which do not raise direct societal risks. Advancing multimodal reasoning has strong potential for positive impact in domains like healthcare, accessibility, and education. Rigorous benchmarks like $\mathbb{M}$ARBLE can help ensure that future systems are robust and beneficial ahead of deployment.

## C    RELATED WORK

**Chain-of-Thought and multimodal reasoning paradigms.**    The Chain-of-Thought (CoT) prompting paradigm has significantly advanced reasoning in language models by enabling stepwise decomposition of complex problems (Wei et al., 2022). The Multimodal Chain-of-Thought (MCoT), its extension to the multimodal domain, represents a natural progression, encouraging models to articulate intermediate reasoning steps while integrating multiple modalities such as images, text, and diagrams. Recent works like Wang et al. (2025a) highlight prompt-based, plan-based, and learning-based MCoT strategies, yet also underscore the lack of robust, diagnostic benchmarks tailored to multimodal reasoning.

Recent multimodal instruction tuning approaches fine-tune LLMs augmented with visual encoders to follow multimodal prompts (Li et al., 2024; Zhu et al.). While these models can generate fluent outputs, their reasoning often lacks depth or consistency, particularly on tasks involving spatial, numerical, or abstract visual patterns (Yue et al., 2024; Chia et al., 2024).

**Multimodal reasoning benchmarks.**    Several datasets have been proposed to evaluate multimodal reasoning, such as ScienceQA (Lu et al., 2022), MMMU (Yue et al., 2024), MathVista (Lu et al., 2023a), EMMA Hao et al. (2025) and MEGABench (Chen et al., 2024). These benchmarks span academic knowledge domains and require integrating visual and textual information. However, they often prioritize answer accuracy over the evaluation of the full reasoning trace, making it difficult to diagnose model errors. Others, like PuzzleVQA (Chia et al., 2024) and NLVR (Wu et al., 2024), introduce abstract reasoning challenges but are limited in modality diversity and stepwise supervision. Recent works like Critic-V Zhang et al. (2025) and MMIR Yan et al. (2025) introduced frameworks for multimodal inconsistency detection or critic-guided refinement, which improved performance but was limited to rather shallow reasoning paths.

There are few previous benchmarking approaches that leveraged multimodal tasks inspired by video game puzzle environments (Zheng et al., 2025; Paglieri et al., 2024; Topsakal et al., 2024). Most recently and closely related, Wang et al. (2025b) proposed MM-Escape, an escape-room like environment where MLLMs have to navigate and leverage the surroundings (*e.g.,* retrieving a hidden key) in order to escape a room. While this benchmark shares some similarity with the $\mathbb{M}$-PORTAL task in $\mathbb{M}$ARBLE, $\mathbb{M}$-PORTAL introduces a novel and much harder, multi-step problem solving challenge. To illustrate this, consider GPT-4o model which solved $70 - 100\%$ of the maps in MM-Escape, but performed very poorly on $\mathbb{M}$-PORTAL (*e.g.*, $4.1\%$ accuracy on `fill-the-blanks`).

# D    ILLUSTRATION OF EXAMPLE PROBLEMS

## D.1    𝕄-PORTAL

### Portal 2: Complex multi-step problem solving

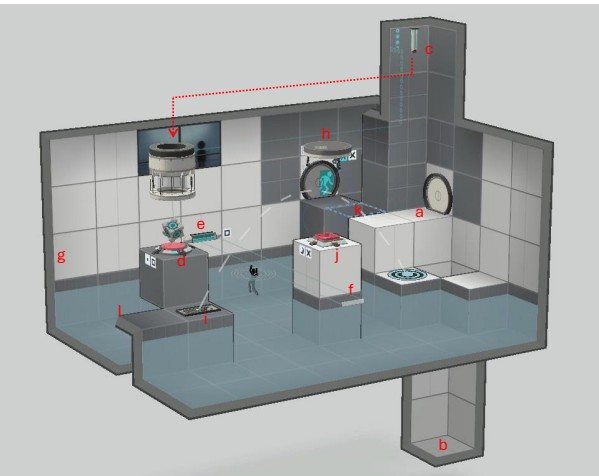

### Solution:

Step 1: Place portals in positions a, b and jump down into b to get ejected from a to press the button c.

Step 2: Button c releases a cube to land on button d which activates the bridge e.

Step 3: Place portals in positions f, g to walk across the bridge towards the cube at location d.

Step 4: Pick up the cube and step on button d which also activates the downwards pushing tractor beam at location h.

Step 5: Throw the cube down to the device at i that catapults it over to the target area.

Step 6: The tractor beam intercepts the cube and pushes it on the slot j which opens the (blue) exit door and elevates a platform at location k.

Step 7: Place portals in positions l, a, walk through l, walk across k to reach the exit.

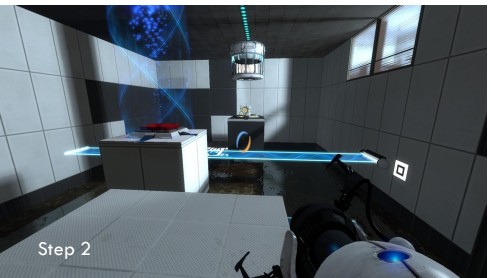

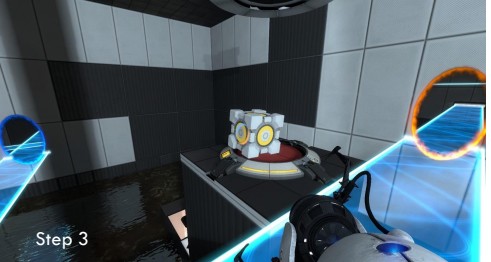

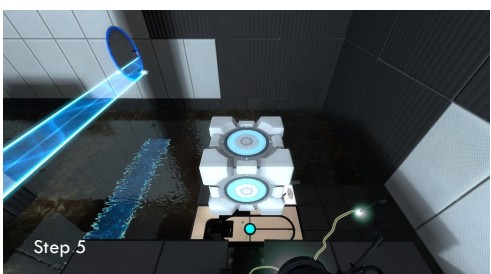

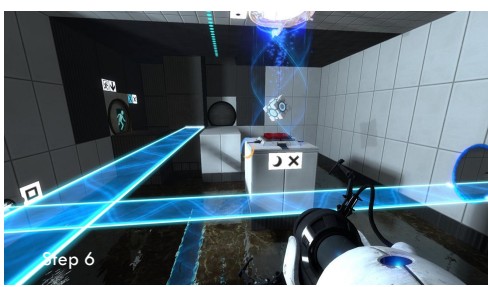

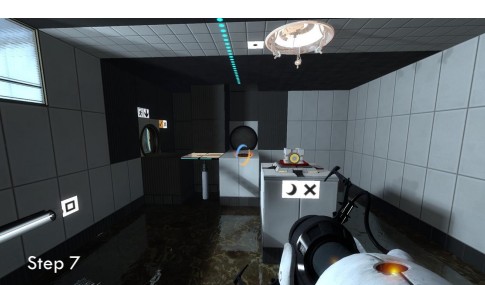

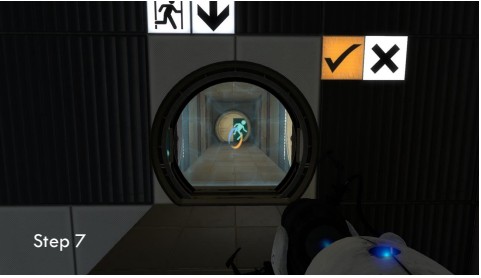

Figure 8: Overview of the Portal-2 Dataset of the MARBLE-Benchmark. Illustrated is a rather basic level Portal 2 problem, which only requires seven steps to solve. For comparison, the advanced problems introduced in this benchmark may involve several dozens of steps. Also, steps are not always decomposed into their most atomic form to keep enough complexity within a step to make mistaken steps harder to detect.

**Problem images** (excerpt)

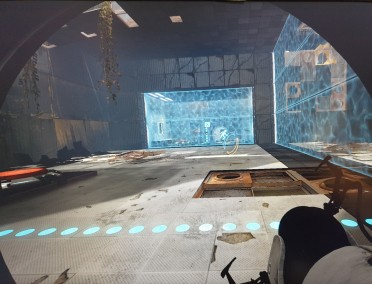

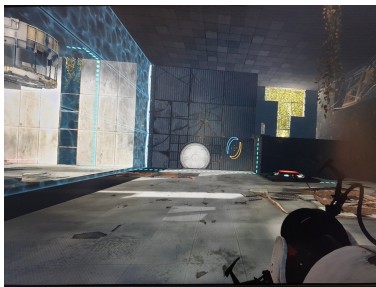

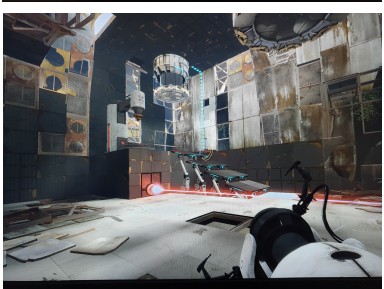

**Hint image**

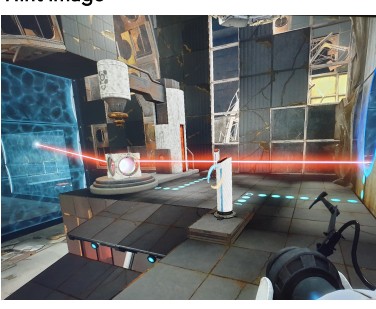

**Problem description**

"You enter room 1, which is connected to room 2 on the right, separated by a shield wall. Room 1 contains a button on the floor that activates a stair leading up to a platform. On this platform, there is a switch that controls a mirror cube machine located in room 2. Room 2 features a laser source that hits the wall and a laser teleportation machine. When activated by a button press, this teleportation machine sends any object placed on it (such as a cube) to the endpoint of the laser ray, wherever the laser is directed. This allows cubes to travel through shield walls that would otherwise block movement. However, teleportation does not work through solid walls. Room 2 also has a button that activates a cube machine located next to the teleportation device. Room 3 is separated from room 1 by a shield wall and contains a button that opens the door to room 4. Room 4 is a small area with only a button on the floor, which opens the exit door."

**Solution** 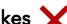

"**Step 1**: Go to room 2 (on the right) and press the switch to drop a cube.",

"**Step 2**: Shoot a blue portal where the laser hits the wall and one on the wall that points to the central room (room 1).",

"**Step 3**: Place the cube on the laser teleportation machine and press the switch to send the cube via laser to room 1.",

"**Step 4**: Go to room 1 and place the cube on the button.",

"**Step 5**: Walk up the stairs to press the little button, which drops a mirror cube in room 1.",

"**Step 6**: Pick up the mirror cube and place it in front of the laser source such that the laser points towards room 3.",

"**Step 7**: Create a new cube by pressing the little button in room 2.",

"**Step 8**: Place the new cube on the laser teleportation machine and press the button to send the cube.",

"**Step 9**: Pick up the mirror cube and place it on the teleportation device.",

"**Step 10**: Shoot an orange portal where the laser source hits the wall and a blue portal at the wall next to the teleportation device to direct the laser to the mirror cube which needs to point to room 3.",

"**Step 11**: Activate the teleportation machine by pressing the button next to the machine.",

"**Step 12**: Go to room 3, pick one cube, and place it on the button to open the door to room 4. Take the other cube and bring it to room 4, placing it on the button on the floor to open the exit door.",

"**Step 13**: Go through the exit door. Problem solved."

**Mistakes** 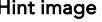

"**Step 2**: Shoot a blue portal where the laser hits the wall and an orange portal on the same wall close to the boundary to room 1 such that the cube gets sent to room 1.",

"**Step 5**: Go to room 2 and collect the mirror cube who dropped due to the button press in room 1.",

"**Step 6**: Pick up the mirror cube and place it in front of the laser source such that the laser points towards room 2.",

"**Step 10**: Shoot an orange portal where the laser source hits the wall and a blue portal at the wall of the entrance in room 1, such that the laser points to room 3.",

"**Step 12**: Go to room 3, pick one cube, and place it on the button of room 4 to open the door in room 4. Take the other cube and placing it on the button of room 3, now both doors are open."

Figure 9: Illustration of an example problem of the $\mathbb{M}$-PORTAL dataset (problem 5), composed of a problem description, images, solution steps, mistakes, and optional hint images.

Figure 8 gives an extended overview of the $\mathbb{M}$-PORTAL problem. It introduces a simple example problem, created for illustrative purposes and does not cover the full complexity the benchmark. Each map in $\mathbb{M}$-PORTAL requires a sequence of actions to solve, making it a complex multimodal reasoning problem.

Figure 9 shows a challenging example problem of the $\mathbb{M}$-PORTAL task of $\mathbb{M}$ARBLE Figure 9 shows input images and instruction text that describe the problem. A manually curated solution is shown on the right side, together with five mistaken steps, below. A hint image depicts the crucial insight that allows to solve the map.

## D.2 M-Cube

Figure 10 presents a complete example question of M-Cube task, and the solution to the instance with the corresponding 2D and 3D visualization. Figure 11 shows the prompt of the perception task.

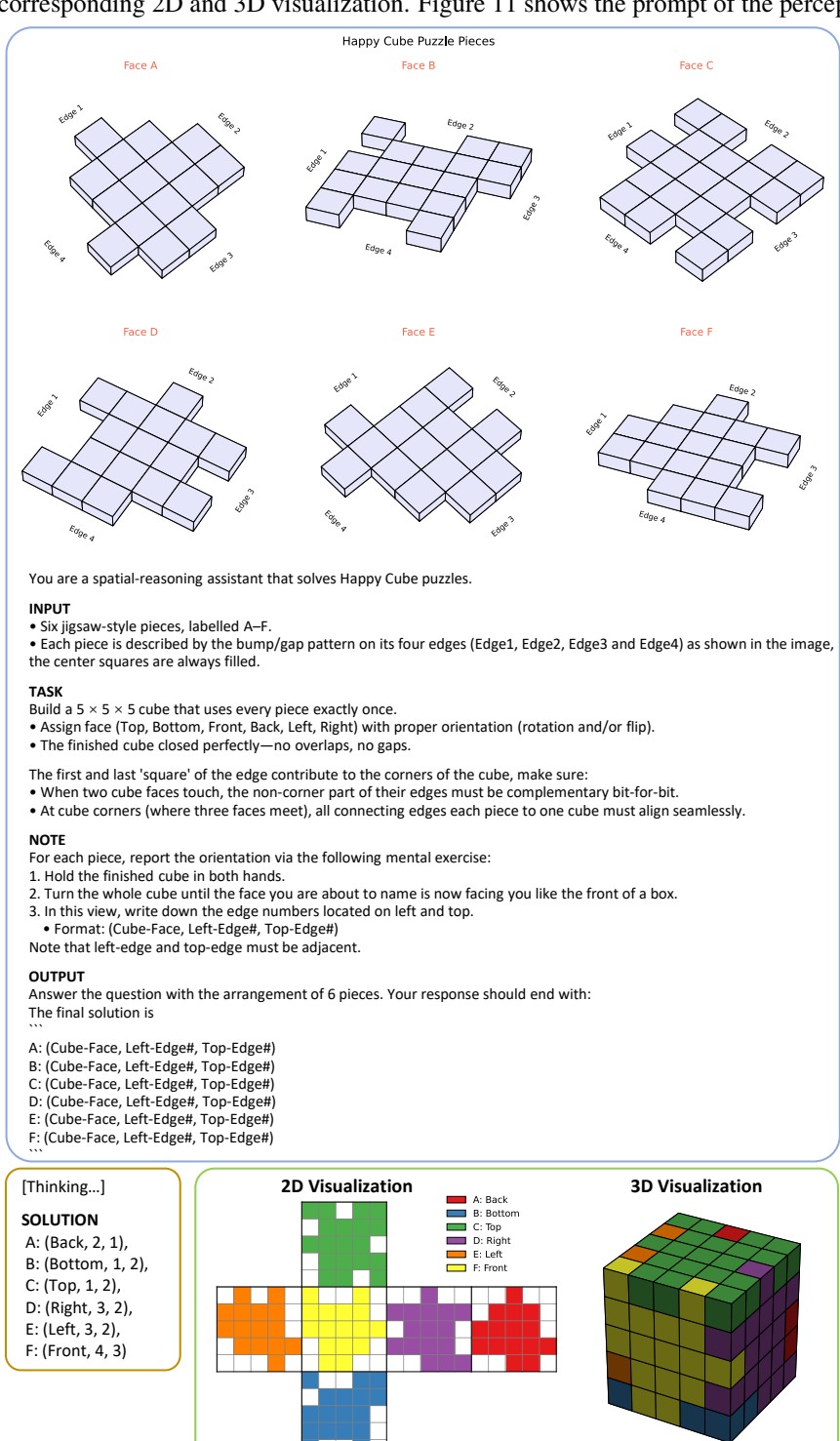

Figure 10: Illustration of M-Cube Problem. *Top*: Example input image and prompt of the problem. *Bottom*: Example solution to the problem (left) and corresponding 2D and 3D visualization (right). The visualization is not part of the inputs or outputs of the benchmark.

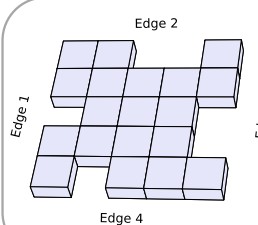

You are given an image of a 5 x 5 grid. In the grid, each cell on the edges is randomly preserved or dropped, while the center 3 x 3 region is always preserved. Now convert the input image into a 2D array, where 0 = gap and 1 = bump, and ensure edge1 = left, edge2 = top, edge3 = right, edge4 = bottom in the 2D array. You should answer with "Here is the converted 2D array: [array]" where [array] is a 2D array in the format of Python list of lists.

Figure 11: Prompt for evaluating the perception ability of MLLMs on 𝕄-CUBE.

The solution validator of 𝕄-CUBE can serve as an auxiliary tool to assist MLLM in solving the reasoning problems. Given a candidate solution, the solution validator could determine whether the solution is correct or not (binary feedback). In addition, it can also provide diagnostic information such as edge conflicts (detailed feedback). Figure 12 illustrates an example where the MLLM leverages feedback from the validator to iteratively refine its solution.

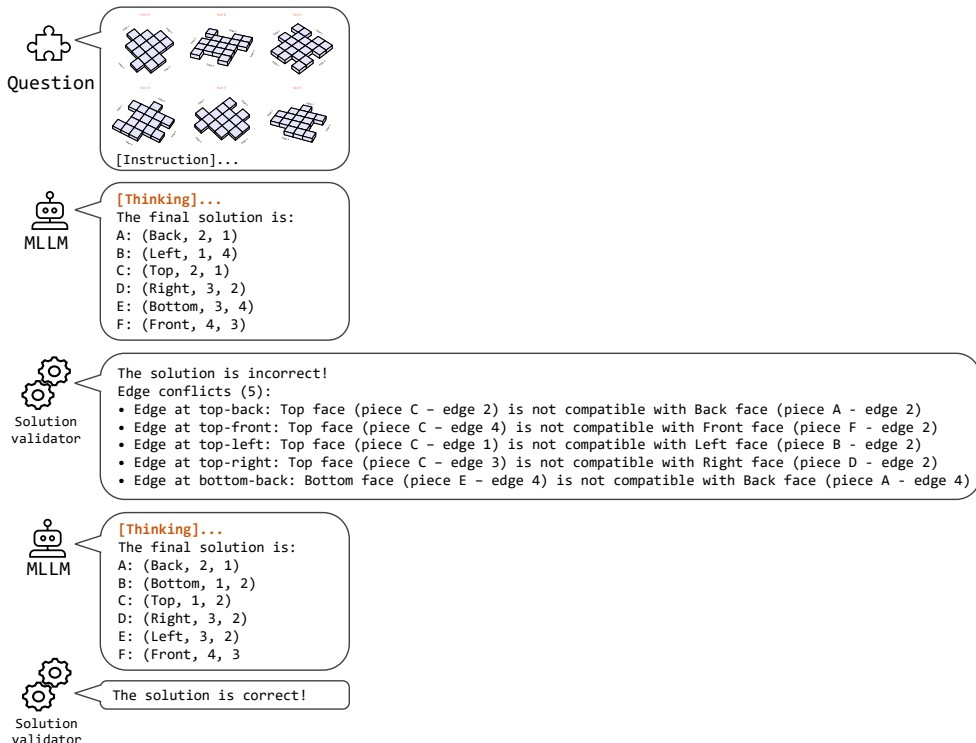

Figure 12: Example of MLLM using solution validator as a tool to gather feedback and iteratively refine its response on the 𝕄-CUBE dataset.

**Results with solution validator.** The ability to use tools or perform function calls has emerged as a crucial feature in latest MLLMs Schick et al. (2023). In case of 𝕄-CUBE, the solution validator could serve as an auxiliary tool to assist MLLMs in tackling complex reasoning tasks. In each round, the model proposes a candidate solution and evaluates it with the solution validator. Based on the validator's feedback, the model could iteratively refine its response towards a better solution in the next round. Specifically, we design two types of feedback: *(i)* Binary feedback, which simply indicates whether a solution is correct or not in a black box manner, *(ii)* Detailed feedback, which not only verifies the correctness of the solution but also provides diagnostic information such as which edges of the cube are in conflict. Figure 13 shows the performance of GPT-o4-mini under both types of feedback. On CUBE-easy, the performance increases significantly for both binary and detailed feedback and detailed feedback consistently outperforms binary feedback, increasing

the performance from 10% to up to 28% accuracy after 5 rounds of interactions, which indicates the value of diagnostic information. However, on more challenging CUBE dataset, the performance using the solution validator tool remains 0% regardless of the feedback type, highlighting the limitation of current MLLMs in solving harder multimodal reasoning problems.

In summary, we introduce a multi-step setup within $\mathbb{M}$-CUBE that enables iterative refinement through the feedback from a solution validator. This setup closely mirrors how humans tackles real-world problem-by making initial attempts, gathering feedback from the environment, and refining their strategies accordingly. However, many current reasoning models would not retain and build upon previous reasoning steps, often discarding the reasoning in earlier context[1], resulting in less effective reasoning in multi-round setup. Therefore, future models capable of interleaved thinking and tool use would benefit more from such validator-assisted setup.

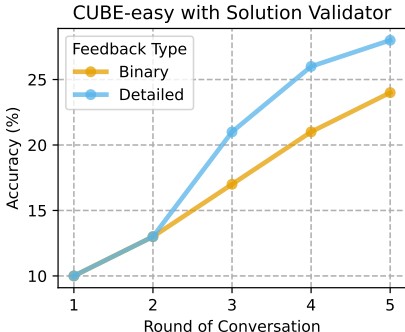

Figure 13: Performance of GPT-o4-mini on CUBE-easy with binary or detailed feedback from solution validator. On CUBE, the performance will remain 0%.

---

[1] Check this OpenAI API document for example.

### D.3 M-MAZE

Figure 14 presents an example question of M-MAZE.

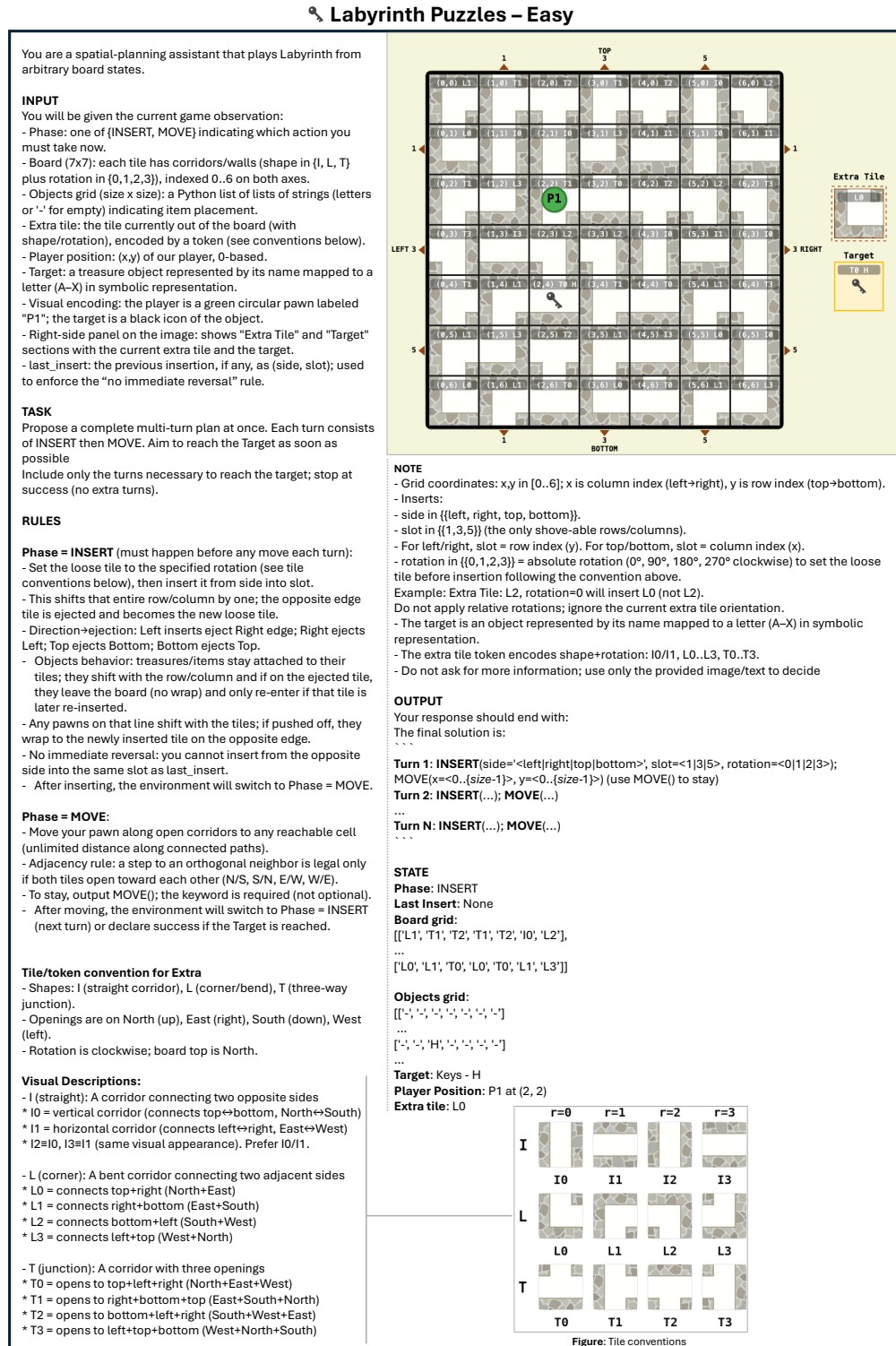

Figure 14: Illustration of M-MAZE Problem: Example input image and prompt of the problem in Visual Harness + Symbolic Representation setting.

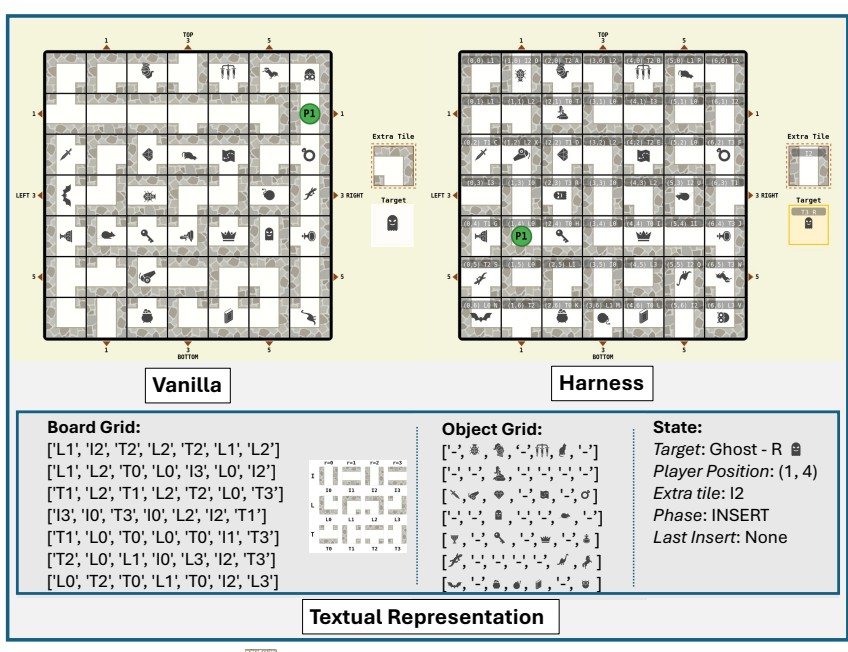

Figure 15: Overview of the 𝕄-MAZE board representation. **Visual Only** corresponds to the *Vanilla* setting where only the board image along selected textual game informations (*Phase*, *target*) are given. **Visual Harness** adds an overlay of the coordinates on each tile (with respect to the Vanilla setting). **Symbolic/Textual representation** adds the *board grid* and *object grid* as arrays in the prompt thereby reducing the required visual parsing capabilities required by the models.

# E    EXPERIMENT DETAILS.

Table 3 provides a comprehensive list of all the models evaluated oin this paper, along with the hyperparameters. We use the same hyperparameters for evaluating both the $\mathbb{M}$-PORTAL and $\mathbb{M}$-CUBE tasks. For open-source models such as Qwen2.5-VL-72B, InternVL3-78B and Llama-4-Scout, we use vLLM Kwon et al. (2023) for efficient inference, with a setting of temperature of 0 and maximum output token length of 16, 000 for all the models. The open-source models are evaluated on the whole evaluation suite of $\mathbb{M}$-CUBE and $\mathbb{M}$-PORTAL.

In contrast, close-source models such as GPT-4o, Claude-3.7-Sonnet, Gemini-2.5-pro, GPT-o3 and GPT-4o-mini are evaluated with their respective APIs. The "reasoning effort" parameter, which controls the allowed length of reasoning chain, is set to "medium" for GPT-4o-mini and Gemini-2.5-Pro, and 12,000 for Claude-3.7 Sonnet. Due to the limit of budget, we choose 200 representative examples on $\mathbb{M}$-CUBE and $\mathbb{M}$-MAZE. The whole set of $\mathbb{M}$-PORTAL is used for evaluating close-source models.

Table 3: MLLMs and corresponding hyperparameters for evaluating $\mathbb{M}$ARBLE benchmark. "Reasoning effort" represents the budget of reasoning tokens to generate before the final response. [*] For reasoning models, max tokens denote the sum of tokens generated for reasoning and final response.

| Model | Date | Temperature | Reasoning Effort | Max Tokens[*] |
|---|---|---|---|---|
| Qwen2.5-VL-72B | 2025.02.19 | 0.0 | - | 16,000 |
| InternVL3-78B | 2025.04.11 | 0.0 | - | 16,000 |
| Llama-4-Scout | 2025.04.05 | 0.0 | - | 16,000 |
| Qwen3-235B-A22B | 2025.04.29 | 0.6 | - | 16,000 |
| GPT-4o | 2024.08.06 | 0.0 | - | 16,000 |
| DeepSeek-R1 | 2025.01.22 | - | - | 16,000 |
| DeepSeek-R1-0528 | 2025.05.28 | - | - | 16,000 |
| Seed-1.5-VL | 2025.04.28 | - | - | 16,000 |
| Claude-3.7-Sonnet | 2025.02.19 | - | 12,000 | 16,000 |
| Gemini-2.5-pro | 2025.05.06 | - | medium | 25,000 |
| GPT-o4-mini | 2025.04.16 | - | medium | 25,000 |
| GPT-o3 | 2025.04.16 | - | medium | 40,000 |
| GPT-5 | 2025.08.07 | - | medium | 40,000 |
| Grok 4 | 2025.07.09 | 0.0 | - | 25,000 |

# F ADDITIONAL RESULTS

## F.1 COST/TOKEN USAGE

| Model | Input $ / 1M | Output $ / 1M | ⚡ M-PORTAL | | | 🧊 M-CUBE | | | 🔲 M-MAZE | | |
|---|---|---|---|---|---|---|---|---|---|---|---|
| | | | In | Out | Total $ | In | Out | Total $ | In | Out | Total $ |
| *OpenAI* | | | | | | | | | | | |
| 🌀 GPT–5 | 1.25 | 10.00 | 0.74 | 14.97 | 15.71 | 0.33 | 4.99 | 5.32 | 0.46 | 12.98 | 13.44 |
| 🌀 o4–mini | 1.10 | 4.40 | 0.90 | 1.36 | 2.26 | 0.66 | 1.38 | 2.04 | 0.69 | 6.31 | 7.00 |
| 🌀 o3 | 2.00 | 8.00 | 1.24 | 5.77 | 7.00 | 0.56 | 2.97 | 3.52 | 0.76 | 16.10 | 16.86 |
| *Anthropic* | | | | | | | | | | | |
| ✳ Claude 3.7 Sonnet | 3.00 | 15.00 | 2.16 | 5.76 | 7.92 | 1.23 | 39.52 | 40.74 | 1.92 | 36.92 | 38.84 |
| *Google* | | | | | | | | | | | |
| ✦ Gemini 2.5 Pro | 1.25 | 10.00 | 0.85 | 1.77 | 2.63 | 0.18 | 16.77 | 16.95 | 0.40 | 29.96 | 30.36 |
| ✦ Gemini 2.5 Flash | 0.30 | 2.50 | 0.21 | 2.21 | 2.42 | 0.04 | 3.96 | 4.00 | 0.10 | 6.22 | 6.32 |
| *DeepSeek* | | | | | | | | | | | |
| 🐋 DeepSeek–R1–250528 | 0.56 | 2.25 | 0.12 | 5.08 | 5.20 | 0.10 | 9.59 | 9.69 | 0.19 | 7.11 | 7.30 |
| *Seed / Doubao* | | | | | | | | | | | |
| 📊 Seed 1.5 VL | 0.42 | 1.26 | 0.29 | 0.39 | 0.68 | 0.15 | 0.98 | 1.13 | 0.22 | 1.13 | 1.35 |
| *xAI* | | | | | | | | | | | |
| 𝕏 Grok–4 | 3.00 | 15.00 | 2.03 | 78.32 | 80.35 | 1.33 | 41.14 | 42.46 | 1.92 | 60.66 | 62.58 |
| 𝕏 Grok–4 Fast | 0.20 | 0.50 | 0.15 | 1.57 | 1.71 | 0.03 | 0.22 | 0.25 | 0.10 | 0.34 | 0.44 |

Table 4: Total inference costs by task ($N = 200$).

| Model | ⚡ M-PORTAL | | 🧊 M-CUBE | | 🔲 M-MAZE | |
|---|---|---|---|---|---|---|
| | In | Out | In | Out | In | Out |
| *OpenAI* | | | | | | |
| 🌀 GPT–5 | 2.96 | 7.49 | 1.32 | 2.50 | 1.84 | 6.49 |
| 🌀 o4–mini | 4.09 | 1.55 | 3.00 | 1.57 | 3.14 | 7.17 |
| 🌀 o3 | 3.10 | 3.61 | 1.40 | 1.86 | 1.90 | 10.06 |
| *Anthropic* | | | | | | |
| ✳ Claude 3.7 Sonnet | 3.60 | 1.92 | 2.05 | 13.17 | 3.20 | 12.31 |
| *Google* | | | | | | |
| ✦ Gemini 2.5 Pro | 3.40 | 0.89 | 0.72 | 8.38 | 1.60 | 14.98 |
| ✦ Gemini 2.5 Flash | 3.50 | 4.42 | 0.67 | 7.92 | 1.67 | 12.44 |
| *DeepSeek* | | | | | | |
| 🐋 DeepSeek–R1 | 1.07 | 11.29 | 0.89 | 21.31 | 1.70 | 15.80 |
| *Seed / Doubao* | | | | | | |
| 📊 Seed 1.5 VL | 3.45 | 1.55 | 1.79 | 3.89 | 2.62 | 4.48 |
| *xAI* | | | | | | |
| 𝕏 Grok–4 | 3.38 | 26.11 | 2.22 | 13.71 | 3.20 | 20.22 |
| 𝕏 Grok–4 Fast | 3.75 | 15.70 | 0.75 | 2.20 | 2.50 | 3.40 |

Table 5: Average Token Usage per puzzle (in thousands) per task ($N = 200$).

## F.2 CONFIDENCE INTERVAL OF HUMAN EVALUATION

Table 6: Performance of human (mean ± std) on all the tasks of MARBLE.

| | ⚡ M-PORTAL | | 🧊 M-CUBE | | 🔲 M-MAZE | |
|---|---|---|---|---|---|---|
| Models | Binary | Blanks | CUBE | CUBE-easy | MAZE | MAZE-easy |
| 👤 *Human* | - | $37.5 \pm 22.2$ | $0.0 \pm 0.0$ | $85.0 \pm 12.2$ | $55.0 \pm 7.1$ | $80.0 \pm 14.1$ |

## F.3 PERCEPTION

### F.3.1 BOARD PARSING

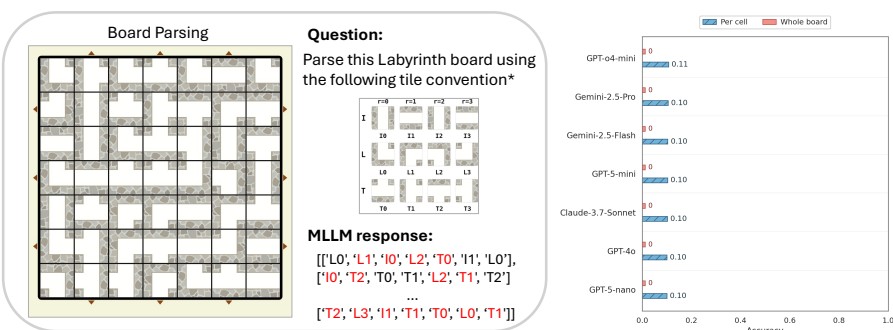

Figure 16: **Perception is also a bottleneck for $\mathbb{M}$-MAZE.** *Left*: Similar to 5, board parsing is a perception task designed to test MLLM's ability on retrieve structured information from visual input and example response of an MLLM. *Right*: Performance of 7 MLLMs on this perception task based on 200 test examples. Accuracy is measured both at individual cells and for the entire $7 \times 7$ board. All the MLLMs perform poorly and completely fail on the whole-board accuracy. *we modify the prompt for readability and avoid redundancy with earlier sections

### F.3.2 GAMESTATE PARSING

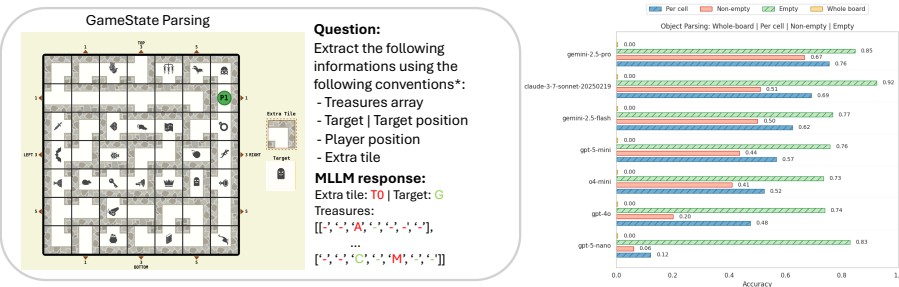

Figure 17: *Left*: GameState parsing task testing MLLMs' ability to extract structured information (treasures array, target and position, player position, extra tile) from visual board input (full prompt in Appendix) and example MLLM response. *Right*: Performance of 7 MLLMs on this perception task based on 200 test examples. Accuracy measured at individual cells (per cell, non-empty, empty) and for the entire board. All MLLMs perform poorly and completely fail on whole-board accuracy. *we modify the prompt for readability and avoid redundancy with earlier sections

### F.3.3 OPAQUE ABLATION

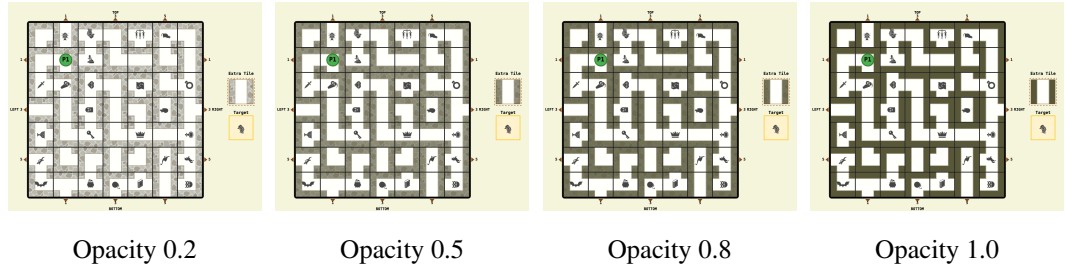

| Opacity 0.2 | Opacity 0.5 | Opacity 0.8 | Opacity 1.0 |

Figure 18: Visual ablation on $\mathbb{M}$-MAZE across varying texture opacity levels.

Table 7: Opacity Sweep on **MAZE** task on **GPT-5-mini** with D=0 and oracle perception (Visual Harness + Symbolic).

| Model | O = 0.0 | O = 0.2 | O = 0.5 | O = 0.8 | O = 1.0 |
|---|---|---|---|---|---|
| ⑤ GPT-5-mini | 0.19 | 0.22 | 0.17 | 0.21 | 0.14 |

## F.4 ISOLATING PERCEPTION FROM PLANNING DIFFICULTY.

### F.4.1 M-CUBE

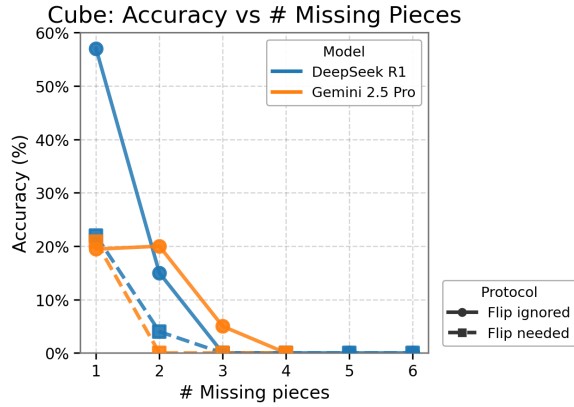

Figure 19: Performance of **DeepSeek R1** & **Gemini 2.5 Pro** across varying levels of task difficulty of the M-CUBE dataset (F=P) | Symbolic representation

### F.4.2 M-MAZE

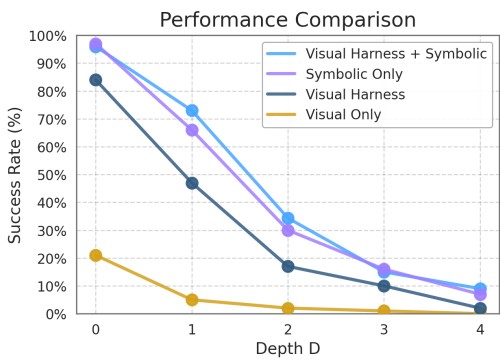

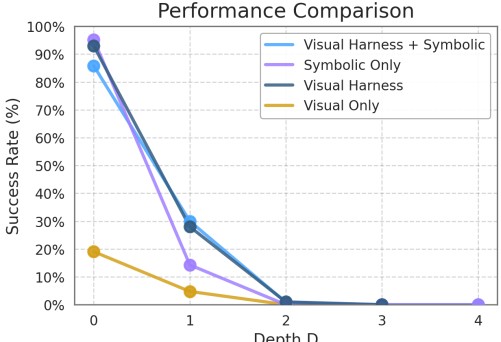

Figure 20: Success rate (%) of GPT-5-mini on MAZE using different visual settings across depths.

Figure 21: Success rate (%) of Gemini 2.5 Flash on MAZE using different visual settings across depths

## F.5 Intermediate Success Metrics.

### F.5.1 🟩 M-Cube

For M-Cube, we have the following additional metrics:

- **Edit distance to GT** ($\downarrow$): Per-sample distance over 6 pieces comparing (face, left-edge, top-edge); range 0–18. Normalized $= \frac{\text{distance}}{18}$.
- **Correct faces** ($\uparrow$): Pieces placed on the correct cube face, ignoring rotation; range 0–6. Ratio $= \frac{\text{value}}{6}$.
- **Rotation correct** ($\uparrow$): Among face-correct pieces, fraction with correct (left-edge, top-edge) orientation.
- **Connectivity violations** ($\downarrow$): Count of adjacent face-edge pairs that fail to interlock when assembled; range 0–12 (12 total adjacencies).

Table 8: 🟩 M-Cube- Intermediate Success Metrics

| Model | Edit Dist. ($\downarrow$) | Correct Faces ($\uparrow$) | Connectivity Violations ($\downarrow$) |
|---|---|---|---|
| ✳ Claude Sonnet 3.7 | 14.00 | 1.18 | 8.62 |
| ☁ DeepSeek R1 | 13.73 | 1.07 | 8.98 |
| 📊 Seed 1.5 VL | 14.07 | 0.95 | 5.54 |
| ✦ Gemini 2.5 Flash | 14.04 | 0.96 | 9.80 |
| ✦ Gemini 2.5 Pro | 14.14 | 0.81 | 10.08 |
| ⑤ GPT 4o | 14.10 | 1.20 | 10.20 |
| ⑤ GPT 5 | 13.78 | 0.78 | 6.89 |
| ✗ Grok 4 0709 | 13.93 | 1.13 | 9.79 |
| ✗ Grok 4 Fast Reasoning | 14.12 | 1.00 | 7.47 |
| 🌊 InternVL3 78B | 14.10 | 0.98 | 1.94 |
| ∞ Llama 4 Scout 17B 16E Instruct | 14.17 | 0.84 | 3.96 |
| ⑤ o3 | 13.69 | 1.18 | 5.46 |
| ⑤ o4-mini | 13.92 | 0.99 | 3.74 |
| 🐋 Qwen2.5 VL 72B Instruct | 13.94 | 1.07 | 9.78 |
| 🐋 Qwen3 235B A22B | 14.07 | 0.95 | 8.26 |

### F.5.2 🔲 M-Maze

For M-Maze, we have the following additional metrics:

- **Turn closeness** ($\uparrow$): At step $t$, defined as $c_t = 1 - \frac{m_t}{m_0}$, where $m_t$ is the minimal turns-to-go from the current state and $m_0$ is that value at the start. We report $\max_t c_t$ across the trajectory, i.e., the best normalized progress toward the goal.

| Model | Turn Closeness (↑) | Plan Length | Turn Distance |
|---|---|---|---|
| ✳ Claude Sonnet 3.7 | $0.22 \pm 0.26$ | $1.52 \pm 1.18$ | $1.77 \pm 0.71$ |
| 🐋 DeepSeek R1 | $0.24 \pm 0.27$ | $1.83 \pm 0.71$ | $1.73 \pm 0.71$ |
| 🐋 DeepSeek V3 | $0.33 \pm 0.38$ | $1.53 \pm 1.06$ | $1.45 \pm 0.90$ |
| Seed 1.5 VL | $0.15 \pm 0.23$ | $1.30 \pm 0.70$ | $1.92 \pm 0.65$ |
| ✦ Gemini 2.5 Flash | $0.14 \pm 0.23$ | $1.07 \pm 0.54$ | $1.90 \pm 0.64$ |
| ✦ Gemini 2.5 Pro | $0.34 \pm 0.39$ | $1.50 \pm 0.59$ | $1.47 \pm 0.94$ |
| 🌀 GPT 4o | $0.14 \pm 0.25$ | $1.61 \pm 0.79$ | $1.90 \pm 0.70$ |
| 🌀 GPT 5 | $0.72 \pm 0.41$ | $1.89 \pm 0.74$ | $0.63 \pm 0.93$ |
| 🌀 GPT 5 mini | $0.64 \pm 0.39$ | $3.71 \pm 1.75$ | $0.99 \pm 0.98$ |
| 🌀 GPT 5 nano | $0.38 \pm 0.35$ | $3.61 \pm 1.78$ | $1.54 \pm 0.86$ |
| 𝕏 Grok 4 0709 | $0.52 \pm 0.48$ | $1.50 \pm 1.39$ | $1.02 \pm 1.01$ |
| 𝕏 Grok 4 Fast Reasoning | $0.82 \pm 0.34$ | $2.54 \pm 1.09$ | $0.41 \pm 0.81$ |
| 🌀 o3 | $0.74 \pm 0.42$ | $2.05 \pm 1.27$ | $0.55 \pm 0.90$ |

Table 9: 𝕄-MAZE Easy - Intermediate Success Metrics

| Model | Turn Closeness (↑) | Plan Length | Turn Distance |
|---|---|---|---|
| ✳ Claude Sonnet 3.7 | $0.27 \pm 0.17$ | $2.15 \pm 1.97$ | $2.86 \pm 0.38$ |
| 🐋 DeepSeek R1 | $0.34 \pm 0.15$ | $2.00 \pm 0.90$ | $2.62 \pm 0.51$ |
| Seed 1.5 VL | $0.27 \pm 0.07$ | $1.18 \pm 0.59$ | $2.92 \pm 0.27$ |
| ✦ Gemini 2.5 Flash | $0.26 \pm 0.11$ | $1.69 \pm 5.72$ | $2.92 \pm 0.27$ |
| ✦ Gemini 2.5 Pro | $0.28 \pm 0.08$ | $1.09 \pm 0.29$ | $2.90 \pm 0.30$ |
| 🌀 GPT 4o | $0.27 \pm 0.06$ | $1.42 \pm 0.55$ | $2.94 \pm 0.24$ |
| 🌀 GPT 5 | $0.29 \pm 0.10$ | $1.72 \pm 1.43$ | $2.88 \pm 0.33$ |
| 🌀 GPT 5 mini | $0.42 \pm 0.23$ | $3.68 \pm 2.10$ | $2.47 \pm 0.81$ |
| 🌀 GPT 5 nano | $0.46 \pm 0.21$ | $4.32 \pm 1.92$ | $2.34 \pm 0.78$ |
| 𝕏 Grok 4 0709 | $0.21 \pm 0.14$ | $1.21 \pm 0.90$ | $2.92 \pm 0.27$ |
| 𝕏 Grok 4 Fast Reasoning | $0.28 \pm 0.08$ | $1.49 \pm 0.59$ | $2.88 \pm 0.33$ |
| 🌀 o3 | $0.20 \pm 0.15$ | $1.00 \pm 0.93$ | $2.93 \pm 0.29$ |
| 🌀 o4-mini | $0.40 \pm 0.20$ | $3.49 \pm 1.37$ | $2.55 \pm 0.77$ |

Table 10: 𝕄-MAZE- Intermediate Success Metrics

## F.6 ERROR ANALYSIS

To strengthen the insights drawn from 𝕄ARBLE, as suggested by reviewers, we categorized model errors across all three tasks. Specifically, we employed a **human-in-the-loop multi-label classification** approach, using experienced players of the respective tasks as annotators, to provide a fine-grained analysis of model capabilities. We report detailed error proportions below. This analysis focuses on **Claude Sonnet 3.7** & **Gemini 2.5 Pro** which expose their reasoning traces.

### F.6.1 🎛 𝕄-MAZE

**Adjacency Error** .

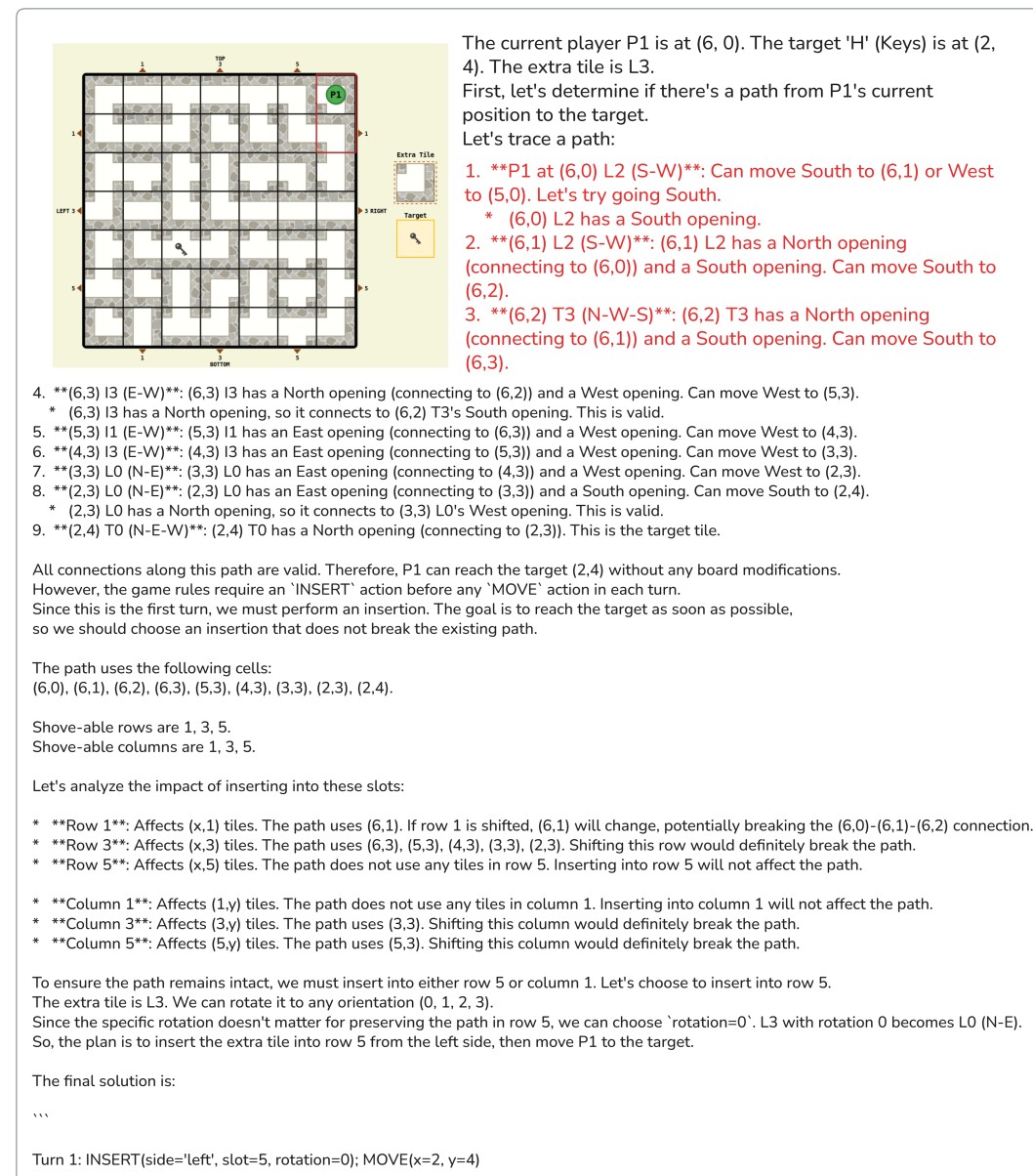

The current player P1 is at (6, 0). The target 'H' (Keys) is at (2, 4). The extra tile is L3.
First, let's determine if there's a path from P1's current position to the target.
Let's trace a path:

1. **P1 at (6,0) L2 (S-W)**: Can move South to (6,1) or West to (5,0). Let's try going South.
   * (6,0) L2 has a South opening.
2. **(6,1) L2 (S-W)**: (6,1) L2 has a North opening (connecting to (6,0)) and a South opening. Can move South to (6,2).
3. **(6,2) T3 (N-W-S)**: (6,2) T3 has a North opening (connecting to (6,1)) and a South opening. Can move South to (6,3).

4. **(6,3) I3 (E-W)**: (6,3) I3 has a North opening (connecting to (6,2)) and a West opening. Can move West to (5,3).
   * (6,3) I3 has a North opening, so it connects to (6,2) T3's South opening. This is valid.
5. **(5,3) I1 (E-W)**: (5,3) I1 has an East opening (connecting to (6,3)) and a West opening. Can move West to (4,3).
6. **(4,3) I3 (E-W)**: (4,3) I3 has an East opening (connecting to (5,3)) and a West opening. Can move West to (3,3).
7. **(3,3) L0 (N-E)**: (3,3) L0 has an East opening (connecting to (4,3)) and a West opening. Can move West to (2,3).
8. **(2,3) L0 (N-E)**: (2,3) L0 has an East opening (connecting to (3,3)) and a South opening. Can move South to (2,4).
   * (2,3) L0 has a North opening, so it connects to (3,3) L0's West opening. This is valid.
9. **(2,4) T0 (N-E-W)**: (2,4) T0 has a North opening (connecting to (2,3)). This is the target tile.

All connections along this path are valid. Therefore, P1 can reach the target (2,4) without any board modifications.
However, the game rules require an `INSERT` action before any `MOVE` action in each turn.
Since this is the first turn, we must perform an insertion. The goal is to reach the target as soon as possible,
so we should choose an insertion that does not break the existing path.

The path uses the following cells:
(6,0), (6,1), (6,2), (6,3), (5,3), (4,3), (3,3), (2,3), (2,4).

Shove-able rows are 1, 3, 5.
Shove-able columns are 1, 3, 5.

Let's analyze the impact of inserting into these slots:

* **Row 1**: Affects (x,1) tiles. The path uses (6,1). If row 1 is shifted, (6,1) will change, potentially breaking the (6,0)-(6,1)-(6,2) connection.
* **Row 3**: Affects (x,3) tiles. The path uses (6,3), (5,3), (4,3), (3,3), (2,3). Shifting this row would definitely break the path.
* **Row 5**: Affects (x,5) tiles. The path does not use any tiles in row 5. Inserting into row 5 will not affect the path.

* **Column 1**: Affects (1,y) tiles. The path does not use any tiles in column 1. Inserting into column 1 will not affect the path.
* **Column 3**: Affects (3,y) tiles. The path uses (3,3). Shifting this column would definitely break the path.
* **Column 5**: Affects (5,y) tiles. The path uses (5,3). Shifting this column would definitely break the path.

To ensure the path remains intact, we must insert into either row 5 or column 1. Let's choose to insert into row 5.
The extra tile is L3. We can rotate it to any orientation (0, 1, 2, 3).
Since the specific rotation doesn't matter for preserving the path in row 5, we can choose `rotation=0`. L3 with rotation 0 becomes L0 (N-E).
So, the plan is to insert the extra tile into row 5 from the left side, then move P1 to the target.

The final solution is:

```

Turn 1: INSERT(side='left', slot=5, rotation=0); MOVE(x=2, y=4)
```

Figure 22: A reasoning trace from ♦ Gemini 2.5 Flash on the 🎛 𝕄-MAZE task, demonstrating a critical **adjacency error**. Although the model attempts to trace a path, it incorrectly assumes that two tiles are connected, failing to properly verify the tile openings. This misinterpretation of the board state leads it to find an invalid path, a common failure mode for models on this task.

**State-Update Error** .

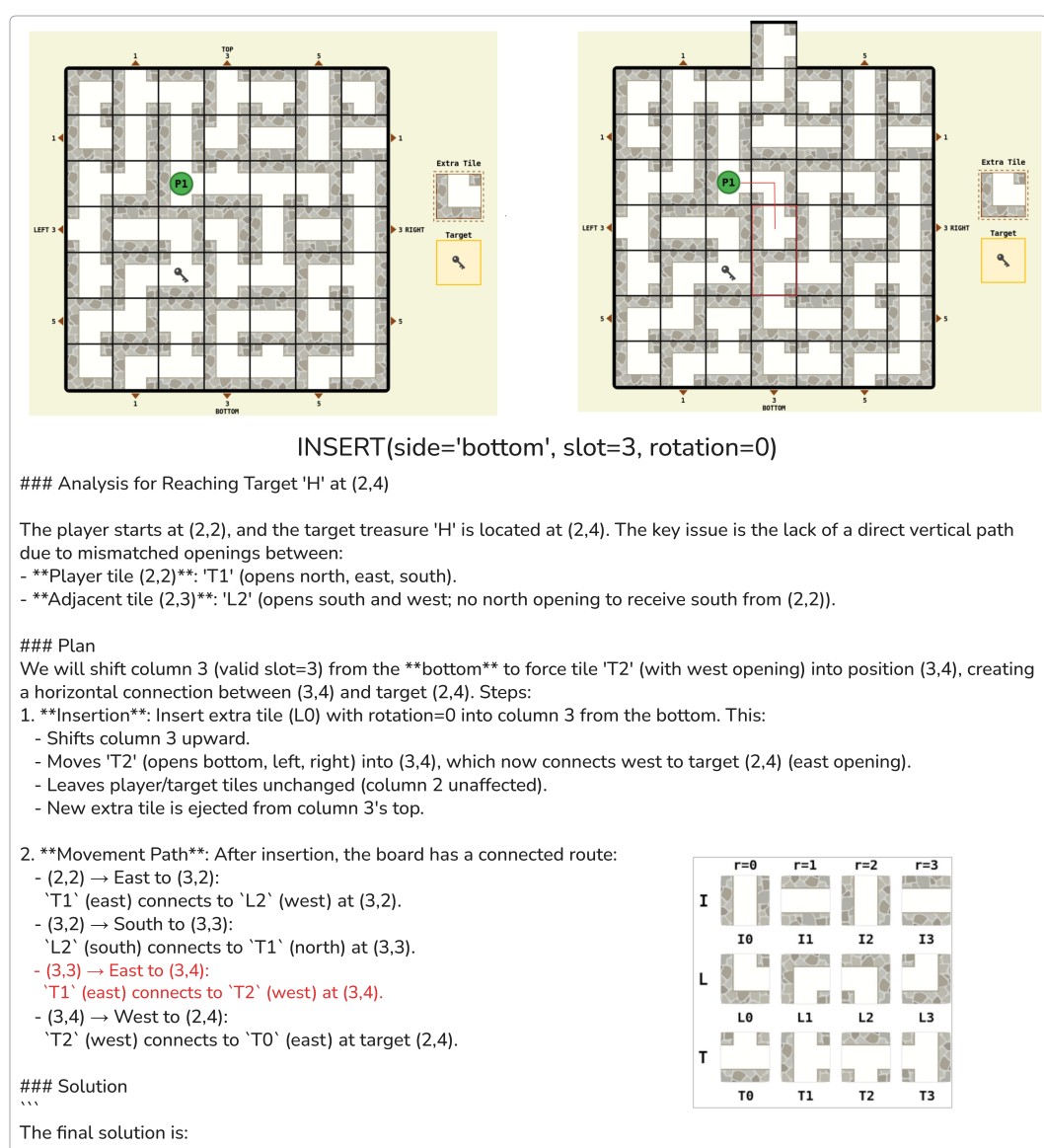

INSERT(side='bottom', slot=3, rotation=0)

### Analysis for Reaching Target 'H' at (2,4)

The player starts at (2,2), and the target treasure 'H' is located at (2,4). The key issue is the lack of a direct vertical path due to mismatched openings between:
- **Player tile (2,2)**: 'T1' (opens north, east, south).
- **Adjacent tile (2,3)**: 'L2' (opens south and west; no north opening to receive south from (2,2)).

### Plan
We will shift column 3 (valid slot=3) from the **bottom** to force tile 'T2' (with west opening) into position (3,4), creating a horizontal connection between (3,4) and target (2,4). Steps:
1. **Insertion**: Insert extra tile (L0) with rotation=0 into column 3 from the bottom. This:
   - Shifts column 3 upward.
   - Moves 'T2' (opens bottom, left, right) into (3,4), which now connects west to target (2,4) (east opening).
   - Leaves player/target tiles unchanged (column 2 unaffected).
   - New extra tile is ejected from column 3's top.

2. **Movement Path**: After insertion, the board has a connected route:
   - (2,2) → East to (3,2):
     `T1` (east) connects to `L2` (west) at (3,2).
   - (3,2) → South to (3,3):
     `L2` (south) connects to `T1` (north) at (3,3).
   - (3,3) → East to (3,4):
     `T1` (east) connects to `T2` (west) at (3,4).
   - (3,4) → West to (2,4):
     `T2` (west) connects to `T0` (east) at target (2,4).

### Solution
```
The final solution is:
Turn 1: INSERT(side='bottom', slot=3, rotation=0); MOVE(2,4)
```

Figure 23: A reasoning trace from 🐋 DeepSeek R1 on the ▦ 𝕄-MAZE task, demonstrating a critical **state-update error**. The model incorrectly processes the row/column shift from an `INSERT` action, failing to update its internal representation of the board state. This leads it to plan a path based on a hallucinated board, a common failure mode for models on this task.

**Legality Errors** .

```
Plan:
Turn 1: INSERT(side='right', slot=5, rotation=0); MOVE(x=5, y=4)
Turn 2: INSERT(side='top', slot=5, rotation=2); MOVE()
Turn 3: INSERT(side='left', slot=3, rotation=1); MOVE(x=4, y=3)
Turn 4: INSERT(side='bottom', slot=3, rotation=0); MOVE(x=3, y=3)
Turn 5: MOVE(x=2, y=3)
Turn 6: MOVE(x=2, y=2)
Turn 7: MOVE(x=2, y=1)
Turn 8: MOVE(x=2, y=0)

The final solution is:
```
Turn 1: INSERT(side='right', slot=5, rotation=0); MOVE(x=5, y=4)
Turn 2: INSERT(side='top', slot=5, rotation=2); MOVE()
Turn 3: INSERT(side='left', slot=3, rotation=1); MOVE(x=4, y=3)
Turn 4: INSERT(side='bottom', slot=3, rotation=0); MOVE(x=3, y=3)
Turn 5: MOVE(x=2, y=3)
Turn 6: MOVE(x=2, y=2)
Turn 7: MOVE(x=2, y=1)
Turn 8: MOVE(x=2, y=0)
```
```

```
The final solution is:
```
  Turn 1: INSERT(side='top', slot=6, rotation=0); MOVE(x=6, y=3)
```
```

Figure 24: Examples of **Insert Legality Errors** on the ⊞ $\mathbb{M}$-MAZE task. The top plan (from 🌐 GPT-5-mini) shows illegal turns (e.g., 5-8) that only contain a `MOVE` action, violating the rule that every turn must have an `INSERT`. The bottom plan (from 🔶 Qwen2.5 VL 72B) shows an `INSERT` using `slot=6`, which is an illegal slot, as only 1, 3, 5 are permitted.

**Shallow Planning** .

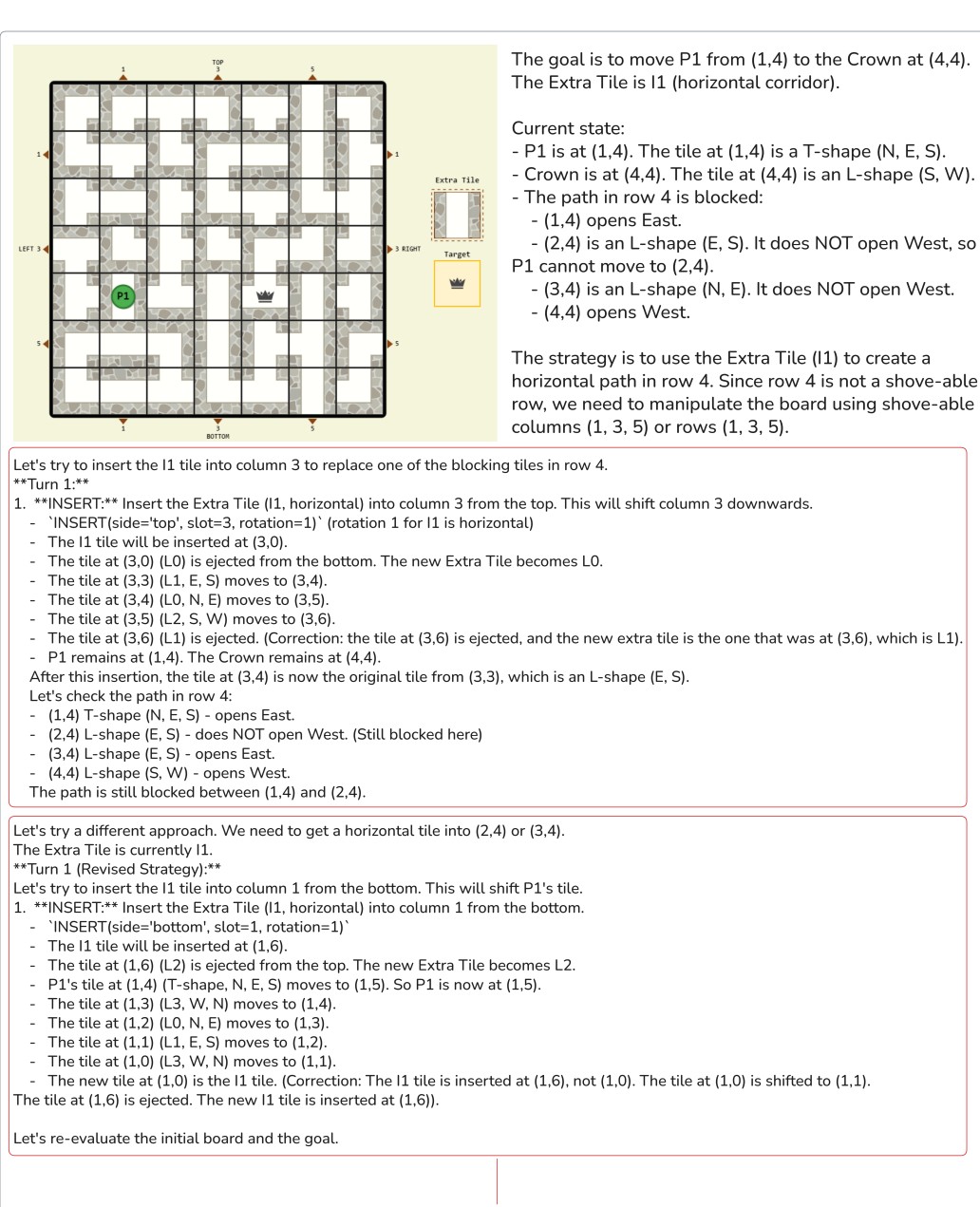

The goal is to move P1 from (1,4) to the Crown at (4,4). The Extra Tile is I1 (horizontal corridor).

Current state:
- P1 is at (1,4). The tile at (1,4) is a T-shape (N, E, S).
- Crown is at (4,4). The tile at (4,4) is an L-shape (S, W).
- The path in row 4 is blocked:
    - (1,4) opens East.
    - (2,4) is an L-shape (E, S). It does NOT open West, so P1 cannot move to (2,4).
    - (3,4) is an L-shape (N, E). It does NOT open West.
    - (4,4) opens West.

The strategy is to use the Extra Tile (I1) to create a horizontal path in row 4. Since row 4 is not a shove-able row, we need to manipulate the board using shove-able columns (1, 3, 5) or rows (1, 3, 5).

Let's try to insert the I1 tile into column 3 to replace one of the blocking tiles in row 4.
**Turn 1:**
1. **INSERT:** Insert the Extra Tile (I1, horizontal) into column 3 from the top. This will shift column 3 downwards.
   - `INSERT(side='top', slot=3, rotation=1)` (rotation 1 for I1 is horizontal)
   - The I1 tile will be inserted at (3,0).
   - The tile at (3,0) (L0) is ejected from the bottom. The new Extra Tile becomes L0.
   - The tile at (3,3) (L1, E, S) moves to (3,4).
   - The tile at (3,4) (L0, N, E) moves to (3,5).
   - The tile at (3,5) (L2, S, W) moves to (3,6).
   - The tile at (3,6) (L1) is ejected. (Correction: the tile at (3,6) is ejected, and the new extra tile is the one that was at (3,6), which is L1).
   - P1 remains at (1,4). The Crown remains at (4,4).
   After this insertion, the tile at (3,4) is now the original tile from (3,3), which is an L-shape (E, S).
   Let's check the path in row 4:
   - (1,4) T-shape (N, E, S) - opens East.
   - (2,4) L-shape (E, S) - does NOT open West. (Still blocked here)
   - (3,4) L-shape (E, S) - opens East.
   - (4,4) L-shape (S, W) - opens West.
   The path is still blocked between (1,4) and (2,4).

Let's try a different approach. We need to get a horizontal tile into (2,4) or (3,4).
The Extra Tile is currently I1.
**Turn 1 (Revised Strategy):**
Let's try to insert the I1 tile into column 1 from the bottom. This will shift P1's tile.
1. **INSERT:** Insert the Extra Tile (I1, horizontal) into column 1 from the bottom.
   - `INSERT(side='bottom', slot=1, rotation=1)`
   - The I1 tile will be inserted at (1,6).
   - The tile at (1,6) (L2) is ejected from the top. The new Extra Tile becomes L2.
   - P1's tile at (1,4) (T-shape, N, E, S) moves to (1,5). So P1 is now at (1,5).
   - The tile at (1,3) (L3, W, N) moves to (1,4).
   - The tile at (1,2) (L0, N, E) moves to (1,3).
   - The tile at (1,1) (L1, E, S) moves to (1,2).
   - The tile at (1,0) (L3, W, N) moves to (1,1).
   - The new tile at (1,0) is the I1 tile. (Correction: The I1 tile is inserted at (1,6), not (1,0). The tile at (1,0) is shifted to (1,1)).
   The tile at (1,6) is ejected. The new I1 tile is inserted at (1,6)).

   Let's re-evaluate the initial board and the goal.

• • •

Figure 25: A reasoning trace from ✦ Gemini 2.5 Flash on the ▦ 𝕄-MAZE task, demonstrating a **shallow planning error**. The model fails to narrow the search space, producing an overly verbose reasoning trace that explores multiple incorrect `INSERT` actions. This exhaustive but aimless exploration continues until the token budget is reached, ultimately resulting in a failure to output any valid plan.

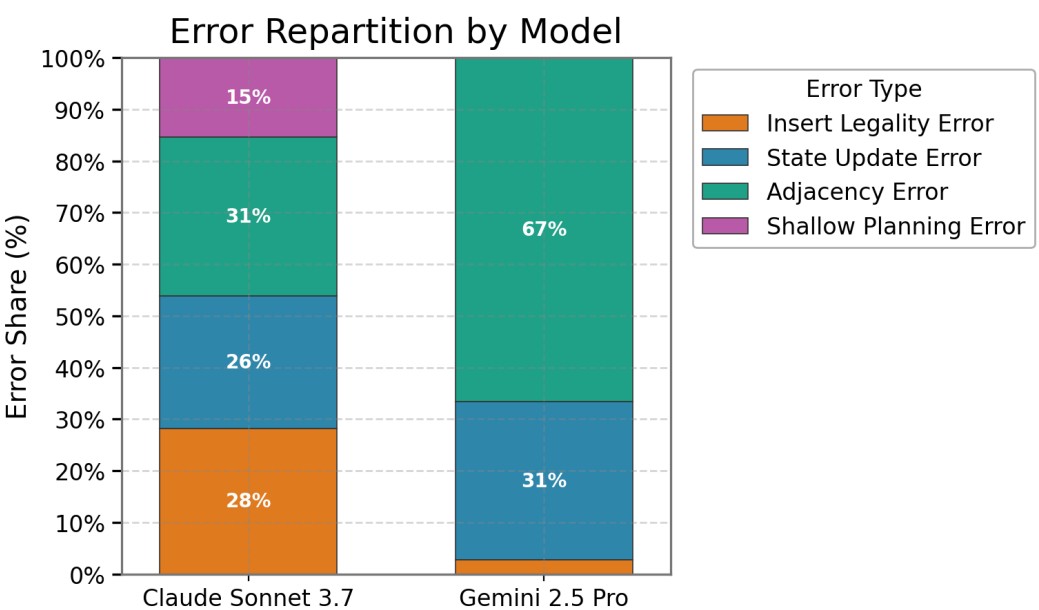

Figure 26: Error composition for Claude Sonnet 3.7 & Gemini 2.5 Pro on the MAZE Easy (Oracle Parsing) task. The stacked bar shows each category's share of all annotated failures (N=25).

