# OpenReview forum: "MARBLE: A Hard Benchmark for Multimodal Spatial Reasoning and Planning"
_ICLR.cc/2026/Conference — Submitted to ICLR 2026_

### Official Review · Reviewer_iioQ · 2025-10-27

**Soundness:** 3
**Presentation:** 3
**Contribution:** 2
**Rating:** 4
**Confidence:** 4

**Summary:**

- This paper present MARBLE, a challenging multimodal reasoning benchmark for MLLMs on multi-step planning and spatial reasoning.
- The motivation for MARBLE is to fill the gap between previous benchmarks for lacking evaluation of step-by-step reasoning in planning and spatial reasoning domain.
- MARBLE consistes of three main tasks, M-Portal, M-Cube, M-Maze, in total of 5,024 samples.
- The evaluation results of MARBLE on 12 leading MLLMs show that most model achieve nearly random or 0% performance on MARBLE except for Grok-4 and GPT-5.
- The authors claim that according to failure analysis, perception abilities are still major bottlenecks in multimodal reasoning.

**Strengths:**

- The motivation of the paper is clear, the paper aims to fill the blank of exisiting multimodal language model benchmarks for lacking the gap of structured reasoning steps
- The paper present detailed descriptions about how to curate the benchmark category by category and what's the meaning of each task
- The data curation pipeline emphysize on the reasoning steps instead of the final outcomes of other benchmarks
- The benchmark contains interaction mechanism with the game applications

**Weaknesses:**

- Most models achieve very limited scores on MARBLE, it is hard to distinguish which model perform better than others. Especially, In Table2, M-Cube dataset, CUBE category, all the models achieve 0 on this. It is confusing to find the meaning of this benchmark if most of the models will fail and have extremely low scores.

- The paper significantly lacks insight about why MLLMs fail on MARBLE but focus too much on the data curation process, making the paper looks a little bit like a technical report instead of a research article.

- The application domain of this paper is poor, the paper only focus on planning and spatial reasoning domain, which is not enough for complex multimodal reasoning. In the meantime, the data source is very limited in single setting, e.g. cube task, in fact, many tasks and data sources can be considered for the spatial reasoning. The distribution of the benchmark is very limited.

- Although the author focus on the data curation pipeline, some details still are not clear because of lacking of necessary visualizations, e.g. Figure1, What exactly are these steps?

**Questions:**

In order to get the authors better prepared for the rebuttal, I propose the following questions:

- L77 missing Table reference, which table are you indicating?
- When designing M-PORTAL and M-CUBE, how did you ensure that the intermediate reasoning traces generated by the model could be accurately evaluated or supervised? Did you consider introducing a human-in-the-loop validation process?
- How do you control for the difficulty of the task itself so it doesn't become the main reason for model failure? For example, when a model fails, is it because its reasoning ability is weak, or because the task setup is too difficult for the model?
- Do the incorrect steps still contribute partially to achieving the task goal, or do they completely lead to failure?
- In the discussion, you mentioned that MARBLE evaluates both the final answer and the reasoning trace. Has any hierarchical evaluation been performed on the model-generated chain-of-thought outputs?
- How's the human performance on your Benchmark?
- Does Chain-of-Thought Prompting improve the overall performance?
- What is the performance improvement of common test-time-scaling methods?
- There are also many existing works on chain-of-thought evaluation, what is your major differences between them?

---

> ### Author Response · Authors · 2025-12-03
>
> We thank the reviewer for the detailed and constructive feedback. We are pleased that the reviewer finds the motivation of MARBLE clear, appreciates the careful task design and category-wise descriptions, and values our focus on reasoning steps and interaction with game environments rather than only final outcomes. We answer the reviewer’s questions and concerns below.
>
> `W1: Most models achieve very limited scores on MARBLE, it is hard to distinguish which model perform better than others.`
>
> We thank the reviewer for the insightful question. In MARBLE, each task contains two subtasks with different difficulty levels. While most models obtain limited scores on the **harder** subtask, **their performance differences become clear on the easier subtask**, allowing meaningful comparison.
>
> For example, as shown in Table 2, open-source models generally lag behind closed-source models. Among the closed-source models, **Grok-4 achieves the best performance on M-Portal**, while **GPT-5 and GPT-o3 perform best on M-CUBE and M-MAZE**, respectively. In addition, Gemini-2.5-Pro and o4-mini also demonstrate highly competitive results. These distinctions show that, despite the overall difficulty of MARBLE, the benchmark still effectively differentiates model capabilities.
>
> `W2: The paper significantly lacks insight about why MLLMs fail on MARBLE but focus too much on the data curation process, making the paper looks a little bit like a technical report instead of a research article.`
>
> We thank the reviewer for the great question. As presented in the manuscript, the major contribution of MARBLE is to present a challenging multimodal reasoning benchmark that exposes the limitation of current MLLMs and drives the development of the models’ capabilities in multimodal reasoning.
>
> First of all, existing multimodal benchmarks are relatively shallow and might be quickly saturated. **MARBLE fills this gap by introducing the challenging tasks** that require models to interpret multimodal structure, plan over physical constraints, and search large state spaces. The evaluation indicates that all the frontier models would still struggle in these tasks.
>
> Besides that, our evaluation of MARBLE also reveals essential limit of current MLLMs:
>  - **Perception bottleneck**. Existing models fails to extract structured information from the input image (Figure 5 and Figure 20)
>  - **Difficulty from large search space**. Current reasoning model struggles to find the correct solution under a huge search space (Figure 5 and Figure 6)
>
> Therefore, we believe that MARBLE provides a diagnostic probe and a research target for future architectures capable of grounded, multi-step multimodal reasoning and planning.
>
> `W3: The application domain of this paper is poor, the paper only focus on planning and spatial reasoning domain, which is not enough for complex multimodal reasoning.`
>
>  - **Spatial reasoning and multi-step planning are foundational** These abilities underpin many core application areas of MLLMs, including robotic manipulation and navigation, autonomous driving and drones, augmented/virtual reality, embodied assistants, and interaction in 3D user interfaces.
>
>  - **Depth Over Breadth** Instead of distributing limited annotation and design capacity across many unrelated settings, MARBLE offers **deep, structured diagnostics** within a focused but important domain. The three tasks target complementary aspects of spatial reasoning (discriminative plan verification, 3D mental rotation/assembly, and dynamic multi-step planning). This design reveals distinct failure modes (e.g., perception vs transformation vs planning), which are not captured by existing broad but shallow benchmarks.
>
> Overall, **MARBLE** is not intended to exhaustively cover all forms of multimodal reasoning, but to provide a targeted, high-resolution benchmark for a core component, visuospatial planning, that is essential for many real-world applications and **is still underrepresented in current evaluations**. We see it as complementary to broader, real-world benchmarks: they assess coverage and robustness across tasks and domains, while **MARBLE** offers fine-grained diagnostics on a critical bottleneck for embodied and interactive MLLM systems.
>
> `W4: Although the author focus on the data curation pipeline, some details still are not clear because of lacking of necessary visualizations, e.g. Figure1, What exactly are these steps? `
>
> We thank the reviewer for this question. In the original Figure 1, each "step" represents a sequence of one or more atomic game actions written by experienced Portal 2 players, such as "Shoot the second portal under the cube, then walk to the platform." All steps are curated to be physically plausible within game mechanics and consistent with map elements (no hallucinated objects), making errors detectable only through understanding spatial consequences.

---

> > ### Author Response · Authors · 2025-12-03
> >
> > `Q1: L77 missing Table reference, which table are you indicating?`
> >
> > L77 refers to Table 1. Thank you for the notification and we’ve updated the manuscript.
> >
> > `Q2: When designing M-PORTAL and M-CUBE, how did you ensure that the intermediate reasoning traces generated by the model could be accurately evaluated or supervised? Did you consider introducing a human-in-the-loop validation process?`
> >
> > For **M-PORTAL**, intermediate plans are generated by an expert human annotator who provides gold-standard Chain-of-Thought solutions and controlled erroneous variants. This enables two closed-form evaluations where the model’s reasoning trace is compared against validated reference steps. Thus, correctness can be assessed precisely without inspecting free-form model outputs.
> >
> > For **M-CUBE** and **M-MAZE**, explicit reasoning supervision is embedded in the task structure: because the search space is combinatorial, a correct final solution is only reachable through consistent intermediate reasoning. Moreover, we provide a programmatic validator and an **online interaction mode** that executes the model’s actions step-by-step and returns environment feedback.
> >
> > We agree that human-in-the-loop could provide more reliable evaluation, which however requires comprehensive human effort and experiences on the tasks. Our current design aims to maximize reliability while keeping evaluation scalable and reproducible without prohibitive annotation cost.
> >
> > `Q3: How do you control for the difficulty of the task itself so it doesn't become the main reason for model failure? For example, when a model fails, is it because its reasoning ability is weak, or because the task setup is too difficult for the model?`
> >
> > Thank you for the important question. We deliberately control the task difficulty from the perception and the search space in the generated tasks. Specifically, we provide textual representation to alleviate the perception issue, and also decrease the search space (by providing partial solutions or controlling the search depth, Figure 4, 5, 6). This design leads to the setting of the easier subtask, where we make sure that the frontier models achieve reasonable and distinguishable performance (as shown in Table 1).
> >
> > `Q4: Do the incorrect steps still contribute partially to achieving the task goal, or do they completely lead to failure?`
> >
> > Following the qualitative error analysis, we argue that this depends on **the ability of the models to self-verify and refine**. Intuitively, constructing a plan on **wrong assumptions will almost invariably lead to task failure**. This is especially observed in the **MAZE** task, even under oracle perception, where **adjacency errors** and **state-update errors** —which stem from incorrect reasoning steps—account for **57%** and **98%** of errors for **Claude Sonnet 3.7** and **Gemini 2.5 Pro**, respectively.
> >
> >  `Q5: Has any hierarchical evaluation been performed on the model-generated chain-of-thought outputs? `
> >
> > We thank the reviewer for this question. Beyond binary success rates, we introduce two forms of hierarchical evaluation over the model-generated plans:
> >  - **Partial success metrics**, which quantify intermediate progress toward the goal (step-/subgoal-level completion) for all three tasks.
> > - **A fine-grained qualitative error analysis** of chain-of-thought traces on the MAZE task, where we organize errors into a structured taxonomy.
> >
> > We provide summaries of these new metrics below and include the full tables in the Quantitative Results section of the Appendix F.5 of the revised manuscript
> >
> > `Q6: How's the human performance on your Benchmark ?`
> >
> > We thank the reviewer for the question. Human performance is provided in Table 2 in the original manuscript. The results indicate that experienced humans could yield competitive performance on the proposed tasks compared to the frontier AI models.
> >
> > `Q7: Does Chain-of-Thought Prompting improve the overall performance ?`
> >
> > Yes, as we’re evaluation the multimodal **reasoning** ability of LLMs, chain-of-thought prompting is by-default used for all the models and plays a crucial role in the problem solving process.
> >
> > `Q8: What is the performance improvement of common test-time-scaling methods?`
> >
> > We thank the reviewer for the interesting question. In the original manuscript, we’ve presented the results of the model performance when scaling the number of attempts in Figure 12 and 13 (Appendix D). The results indicate that the model’s performance increases significantly with increasing test-time compute, for example, CUBE-easy, the model’s performance 10% to up to 28% accuracy after 5 rounds of attempts.

---

### Official Review · Reviewer_14V2 · 2025-10-31

**Soundness:** 3
**Presentation:** 3
**Contribution:** 2
**Rating:** 2
**Confidence:** 4

**Summary:**

This work proposes a suite of tasks designed to evaluate the “think-with-image” capability of multimodal large language models (MLLMs). Specifically, the MARBLE benchmark includes three tasks, i.e., M-Portal, M-Cube, and M-Maze. Each targeting different aspects of multimodal reasoning and spatial planning. Most evaluations are conducted offline, comparing the performance of various MLLMs and providing insightful analyses of their reasoning behaviors.

**Strengths:**

This benchmark is distinctive in its design, incorporating video game environments to construct datasets that evaluate the multimodal reasoning abilities of MLLMs. It features long-horizon tasks with large search spaces, providing a challenging testbed.

The benchmark provides a comprehensive evaluation of various MLLMs, revealing that visual perception and planning remain critical bottlenecks in multimodal reasoning.

**Weaknesses:**

The main focus of this work is on providing a benchmark to evaluate the capabilities of existing MLLMs. It feels largely engineering-oriented, emphasizing data curation rather than introducing new methods to enhance MLLM performance, which limits the conceptual contribution of the paper.

While the overall writing quality is good, it could be made more concise.

The evaluation setup appears somewhat specialized, focusing on a narrow subset of tasks based on simulated images, which may limit the generalizability of the findings.

**Questions:**

Could you provide the image resolution used for each task? For the M-Maze task in particular, if the resolution is too low, tokenized images may lose important fine-grained details that affect spatial reasoning.

Since most tasks are challenging even for humans without prior experience, the major performance gap appears mainly in the maze tasks, while for others, the gap between the best-performing VLMs and humans is relatively small. Although some MLLMs achieve a 0% success rate, how to confidently conclude that perception is the critical bottleneck in multimodal reasoning?

It seems that the difficulty of these tasks also stems from the challenge of remembering intermediate states. The online evaluation results appear more reliable and often better, but only one case was tested this way. Would it be possible to evaluate more models online, or at least explore alternative approaches to approximate online performance?

As these tasks rely on simulated images and specific environments, I am curious about the practical potential or applicability of such benchmarks for assessing real-world MLLM capabilities?

Could you describe the evaluation cost and feasibility of this benchmark for different models?

---

> ### Author Response · Authors · 2025-12-03
>
> We thank the reviewer for the clear and constructive feedback. We are pleased that the reviewer finds MARBLE distinctive in its use of video game environments and long-horizon tasks with large search spaces, and that they appreciate our comprehensive evaluation of many MLLMs and the resulting insights into perception and planning bottlenecks. We answer the reviewer’s questions and concerns below.
>
>
> `W1: The main focus of this work is on providing a benchmark to evaluate the capabilities of existing MLLMs. It feels largely engineering-oriented, emphasizing data curation rather than introducing new methods to enhance MLLM performance, which limits the conceptual contribution of the paper.`
>
> We thank the reviewer for the great question. As presented in the manuscript, the major contribution of MARBLE is to present a challenging multimodal reasoning benchmark that exposes the limitation of current MLLMs and drives the development of the models’ capabilities in multimodal reasoning.
>
> Existing multimodal benchmarks predominantly test shallow perception or retrieval, leaving complex step-by-step reasoning largely unmeasured. These benchmarks are also relatively easy and might be quickly saturated. **MARBLE fills this conceptual gap by introducing the challenging tasks** that require models to interpret multimodal structure, plan over physical constraints, and search large state spaces. Our evaluation indicates that all the frontier models would still struggle in these tasks.
>
> Besides being challenging, MARBLE also reveals essential limit of current MLLMs:
>  - **Perception bottleneck**. Existing models fails to extract structured information from the input image (Figure 5 and Figure 20)
>  - **Difficulty from Large search space**. Current reasoning model struggles to find the correct solution under a huge search space (Figure 5 and Figure 6)
>
> Therefore, we believe that MARBLE provides a diagnostic probe and a research target for future architectures capable of grounded, multi-step multimodal reasoning and planning.
>
>
> `Q1: Could you provide the image resolution used for each task? For the M-Maze task in particular, if the resolution is too low, tokenized images may lose important fine-grained details that affect spatial reasoning. `
>
> Image resolutions vary by task: M-PORTAL: 1920×1080 to 4K (varies by map); M-CUBE: 1800×1000 (uniform); M-MAZE: 1093×896 with board ~500×500px. These resolutions ensure all task-relevant spatial details (portal surfaces, cube orientations, maze tiles) are clearly visible.
>
> `Q2: Since most tasks are challenging even for humans without prior experience, the major performance gap appears mainly in the maze tasks, while for others, the gap between the best-performing VLMs and humans is relatively small. Although some MLLMs achieve a 0% success rate, how to confidently conclude that perception is the critical bottleneck in multimodal reasoning?`
>
> We thank the reviewer for the important question. As indicated in the manuscript, all the model completely fails to correctly extract the information from visual input, resulting in 0% success rate in the perception task. As this is the first step for multimodal reasoning, we therefore conclude that perception remains a major bottleneck for multimodal reasoning.
>
> `Q3: As these tasks rely on simulated images and specific environments, I am curious about the practical potential or applicability of such benchmarks for assessing real-world MLLM capabilities?`
>
> We appreciate the question about real-world applicability and generalizability. While **MARBLE** uses simulated, puzzle-like environments and focuses on a specific family of tasks (visuospatial planning), this is a deliberate methodological choice rather than a limitation.
>
> (i) **Domain-general and foundational skills.** The abilities we target are well-studied, domain-general visuospatial skills in cognitive psychology and are known to transfer across contexts. This is analogous to how abstract math problems are accepted tests of general reasoning even though they are not themselves “real-world” scenes.
>
> (ii) **Links to practical applications.** Each task is designed to mirror concrete application demands in domains where MLLMs are expected to operate: **M-PORTAL** (trajectory + physics) relates to robotics navigation and manipulation in constrained 3D spaces, **M-CUBE** (3D assembly) to manufacturing, construction, and packing/planning scenarios, and **M-MAZE** (dynamic multi-step planning) to autonomous navigation and decision-making in evolving environments. These are precisely the kinds of embodied and interactive settings where integrating spatial reasoning with planning is crucial.
>
> `Q4: Could you describe the evaluation cost and feasibility of this benchmark for different models?`
>
> Thank you for the question. We provide the cost for each task in the section Cost of the Appendix F.1 of the revised manuscript.

---

### Official Review · Reviewer_3WW3 · 2025-11-01

**Soundness:** 3
**Presentation:** 3
**Contribution:** 2
**Rating:** 4
**Confidence:** 4

**Summary:**

The paper introduces MARBLE, a multimodal reasoning benchmark aimed at diagnosing step-by-step spatial reasoning and planning in MLLMs across three tasks: M-PORTAL (Portal-style multi-step planning with visual context), M-CUBE (assembling a 3D cube from six interlocking pieces), and M-MAZE (dynamic 2D labyrinth planning that alternates tile insertion and navigation). Each task has two difficulty settings and emphasizes process-oriented evaluation (e.g., plan correctness, fill-the-blanks, success rate) rather than only final answers. In large-scale evaluations of 12 models, state-of-the-art systems largely fail on the hard settings (≈0% accuracy for M-CUBE and M-MAZE) and only slightly beat random baselines on some M-PORTAL subtasks (with Grok-4 and GPT-5 the best among tested systems), highlighting perception bottlenecks and error accumulation in long-horizon planning. The authors also report an online interactive evaluation loop for M-MAZE that reduces but does not eliminate compounding errors with intermediate feedback.

**Strengths:**

1. The paper is well written and easy to follow.
2. The three tasks probe different mixtures of perception, spatial reasoning, combinatorics, and rule-driven planning. The two-tier difficulty in each task (e.g., CUBE vs CUBE-easy, MAZE vs MAZE-easy) cleanly exposes where models fail (perception vs search vs dynamics).
3. The paper isolates perception with a conversion task (image to edge arrays), showing around 70–76% per-cell accuracy and 0% piece-level accuracy across models, which plausibly explains downstream failures even before reasoning over large search spaces.
4. The action-by-action loop for M-MAZE is practical and closer to agentic usage. It mitigates long-horizon error accumulation, and is helpful for future iterative agents.

**Weaknesses:**

1. Plan-correctness relies on mixing up to five independent mistake steps to produce 2^5 candidates with 1 positive, which is an extreme imbalance that can confound minority-class F1 and encourage shortcut cues in negatives.
2. Human results are reported from 2–3 experienced players; this small N, without variance/confidence intervals, makes it hard to contextualize model gaps especially on hard tasks.
4. While the image to array conversion task is informative, it’s still a single proxy. Consider adding: (a) controlled render sweeps (viewpoints, occlusion, lighting) with sensitivity curves; (b) synthetic text-only surrogates (perfect symbolic inputs) for all tasks to quantify pure reasoning headroom per model; (c) vision-only ablations (frozen LLM head) to profile encoder failure modes; and (d) explicit evaluation of template overfitting in 2D-array formats.
3. Fill-the-blanks and binary plan correctness are valuable, but they do not capture how the plan errs (e.g., spatial consistency violations vs rule misuse vs temporal dependency errors). Consider structured error taxonomies and edit distance between predicted and gold plans (segment-level precision/recall), plus consistency checks (state-update invariants across steps).

**Questions:**

1. How do you ensure that mistaken steps are independent and don’t introduce superficial artifacts (e.g., phrasing patterns) detectable without reasoning?

2. Do you randomize style/syntax across correct vs incorrect steps to avoid annotation style leakage?
3. For each task, can you release paired symbolic inputs (perfect parses) so that one can swap in oracle perception and isolate reasoning gaps by model? How do results change under such oracle perception conditions for all models, not just a subset?
4. What were the token/time budgets per example? Did you enable tool use (e.g., the M-CUBE validator) uniformly across models? If not, could you standardize an evaluation track with validator feedback loops and report Success@k tool calls?
4. For M-PORTAL, how do models behave under viewpoint perturbations, screenshot subsets, or linguistic paraphrases of map instructions? For M-CUBE, can you provide hardness-graded instances (e.g., controlled number of flips, symmetries), and for M-MAZE, depth-controlled splits with matched visual complexity?
5. Will it be possible to add plan-level structured metrics (e.g., step-type confusion, state-consistency violations, illegal-move rate) and per-error attribution dashboards?

---

> ### Author Response · Authors · 2025-12-03
>
> We thank the reviewer for the clear, detailed, and insightful feedback. We are glad that the reviewer finds the paper well written, appreciates that the three tasks probe different mixtures of perception, spatial reasoning, and planning, and values both the perception-isolation experiment and the online M-MAZE loop as informative components of MARBLE. We answer the reviewer’s questions and concerns below.
>
> `W1: Plan correctness, is an extreme imbalance that can confound minority-class F1 and encourage shortcut cues in negatives.`
>
> We thank the reviewer for these related methodological questions. We agree that the plan-correctness subtask involves a highly imbalanced label distribution. However, each candidate plan is evaluated independently without shared context across candidates. Therefore, the model cannot leverage any shortcut cues in the evaluation.
>
>
> `W2: Human results are reported without confidence intervals`
>
> We thank the reviewer for the comment and updated the manuscript to include the confidence intervals of the human performance baseline.
>
> `W3: While the image to array conversion task is informative, it’s still a single proxy. Consider adding: (a) controlled render sweeps (viewpoints, occlusion, lighting) with sensitivity curves; (b) synthetic text-only surrogates (perfect symbolic inputs) for all tasks to quantify pure reasoning headroom per model; `
>
> We thank the reviewer for these suggestions. We've updated the manuscript with the extended experiments:
>  -  **Controlled render perturbations via opacity sweeps**. We evaluate models’ performance on different configurations of opacity (Appendix F.3).
> -  **Isolating perception from planning difficulty**. To better quantify the perception bottleneck and approximate the requested text-only surrogates, we designed ablations that isolate perception difficulty from planning difficulty, allowing us to measure their independent and combined effects on model performance (Appendix F.4).
>
> `Q1: For M-Portal, how do you ensure that mistaken steps are independent and don’t introduce superficial artifacts (e.g., phrasing patterns) detectable without reasoning`
>
> Thank you for the question. All mistaken steps were crafted by **experienced Portal 2 players**. Each mistake preserves causal coherence with later steps and uses varied syntactic structures to avoid linguistic shortcuts. Moreover, each of the five potential mistakes was curated independently by annotators without knowledge of other mistakes in the trajectory, ensuring combinations don't introduce co-occurrence patterns.
>
> `Q2: Do you randomize style/syntax across correct vs incorrect steps to avoid annotation style leakage? `
>
> We use **the same style and syntax** across correct and incorrect steps. For every mistaken step, the difference lies only in the minimal action to the man-present elements (such as pressing button A vs. Button B). This ensures that differences reflect reasoning rather than style cues.
>
> `Q3: For each task, can you release paired symbolic inputs (perfect parses) so that one can swap in oracle perception and isolate reasoning gaps by model? `
>
> Thank you for the great suggestion. We’ll release the paired representation as suggested by the reviewer. In fact, the evaluation of CUBE-easy and MAZE-easy in the manuscript is using the paired representation (Table 2).
>
> `Q4: What were the token/time budgets per example? `
>
> We’ve updated the manuscript to include the average token usage per task in the section Cost/Token Usage of the Appendix F.1.
>
> `Q5: Did you enable tool use (e.g., the M-CUBE validator) uniformly across models?`
>
> For the main experiments, all the models are evaluated without access to the tool-use. To understand the models’ performance in the interactive setting, we conduct experiments with tool-use on selected models and present the results in Figure 7 and Figure 13.
>
>
> `Q6: Will it be possible to add plan-level structured metrics (e.g., step-type confusion, state-consistency violations, illegal-move rate) and per-error attribution dashboards?`
> We thank the reviewer for this insightful suggestion. To address the request for (I) Plan-level structured metrics and (II) Per-Error attribution dashboard, we have computed intermediate success metrics for all tasks and conducted a fine-grained error analysis on the MAZE task. We provide summaries of these new metrics below and added the full tables in the Quantitative Results section of the Appendix (F.5) of the revised manuscript.

---

### Official Review · Reviewer_GE4m · 2025-11-01

**Soundness:** 2
**Presentation:** 2
**Contribution:** 2
**Rating:** 4
**Confidence:** 5

**Summary:**

This paper introduces MARBLE, a benchmark designed to evaluate the ability of multimodal large language models (MLLMs) to perform complex, step-by-step reasoning across modalities. Existing multimodal benchmarks often focus on direct information retrieval from visual inputs or purely text-based reasoning. In contrast, MARBLE targets structured, multi-step planning that integrates spatial, visual, and physical reasoning. The benchmark consists of three tasks—M-Portal, M-Cube, and M-Maze—requiring models to generate and execute multi-step plans in visually grounded environments. Results show that current advanced MLLMs perform poorly on MARBLE, suggesting that complex multimodal reasoning remains an open challenge. The authors further analyze perception bottlenecks and argue that stronger perceptual understanding is essential for progress.

**Strengths:**

Clear and well-structured writing.
The manuscript is clearly written and easy to follow. The benchmark design and evaluation methodology are all presented in a logical, concise manner, supported by informative figures and examples.

Thoughtful dataset and task construction.
The benchmark curation process is technically sound. The detailed pipeline for constructing environments and task sequences provides valuable insight for future multimodal benchmark development.

Comprehensive evaluation and analysis.
The experiments include a broad range of state-of-the-art MLLMs, paired with qualitative analyses of error modes. This thorough evaluation allows the community to clearly understand current limitations and bottlenecks, particularly in perception-driven reasoning.

**Weaknesses:**

Unclear motivation behind task combination.
While each task individually provides meaningful evaluation, the rationale for combining M-Portal, M-Cube, and M-Maze into a single benchmark is not fully articulated. These settings are fairly distinct in format and objective, and similar concepts have appeared in prior embodied and spatial reasoning benchmarks. The paper would benefit from a stronger justification for why combining these tasks yields emergent value beyond scaling and aggregation.

Additional guidance for future research could enhance impact.
Although the paper identifies perception as a bottleneck, it would be helpful to provide more actionable suggestions or directions—e.g., benchmark variants isolating perception versus planning, curriculum strategies, or evaluation under improved perception modules.

In general, introducing yet another challenging benchmark should be justified by strong motivation and a clearly demonstrated gap in existing resources. Without compelling reasons or urgent practical need, adding a new benchmark risks contributing to benchmark inflation rather than advancing the field.

**Questions:**

none

---

> ### Author Response · Authors · 2025-12-03
>
> We thank the reviewer for the thoughtful and valuable feedback assessment to our work. We are happy that the reviewer finds the writing clear, the task and dataset construction technically sound, and the evaluation comprehensive in exposing current perception-driven bottlenecks in MLLMs. We answer the reviewer’s questions and concerns below.
>
>
> `W1: Unclear motivation behind task combination. While each task individually provides meaningful evaluation, the rationale for combining M-Portal, M-Cube, and M-Maze into a single benchmark is not fully articulated.`
>
> We thank the reviewer for the insightful question. We intentionally combine M-PORTAL, M-CUBE, and M-MAZE because each task probes a *different, complementary dimension of multimodal spatial reasoning*:
>  1. **M-PORTAL (long-horizon 3D action planning)**: Tests whether models can maintain spatial coherence when planning action sequences.
>  2. **M-CUBE (3D construction planning)**: Tests mental rotation and collision prediction across assembly steps.
>  3. **M-MAZE (2D dynamic planning)**: Tests dynamic path planning requiring a combination of 2D navigation and environmental state updates.
> Together, all three tasks provide a *holistic* benchmark on the multimodal spatial reasoning abilities of MLLMs under complex scenarios with spatial, visual, and physical constraints.
>
>
> `W2: Although the paper identifies perception as a bottleneck, it would be helpful to provide more actionable suggestions or directions—e.g., benchmark variants isolating perception versus planning, curriculum strategies, or evaluation under improved perception modules.`
>
> We thank the reviewer for this suggestion. To make the perception bottleneck and its implications more actionable, we introduce matched perception/planning variants for all three tasks (summarized in Perception/Planning – Table 1, with results in Tables 2–4). Perception conditions range from raw visual input to near-oracle symbolic surrogates, while planning complexity increases the search space (more blanks/pieces or deeper solutions). Across tasks, models often perform well under symbolic inputs but degrade sharply once they must parse the visual boards, especially on CUBE and MAZE. At the same time, success still decreases as planning difficulty increases even under perfect perception, indicating genuine reasoning limits beyond perception. These variants thus provide concrete evaluation settings where improved perception modules (e.g., external parsers or stronger vision backbones) can be plugged in and compared under fixed planning difficulty.
>
> `W3: In general, introducing yet another challenging benchmark should be justified by strong motivation and a clearly demonstrated gap in existing resources. Without compelling reasons or urgent practical need, adding a new benchmark risks contributing to benchmark inflation rather than advancing the field.`
>
> We thank the reviewer for the insightful question. We fully agree that a new benchmark should be designed to advance the field. This is in fact the principle that guides the design of MARBLE.
>
> First, **many influential reasoning benchmarks began with near-zero model performance**. For quite some time ARC benchmark was really hard for all models, current ARC-AGI benchmarks are still hard even for latest models - in medical domain, MedQA-USmle used to be really hard for years (performance around 30-40% whereas random is 25%) and all of a sudden LLMs became better at it. In contrast, today’s multimodal reasoning benchmarks tend to be relatively shallow for frontier models. **Therefore, it’s important to have a multimodal benchmark that is not quickly saturated.**
>
> Based on this motivation, **MARBLE is intentionally designed as a hard benchmark** that plays two essential roles:
>  - **Revealing systematic capability gaps.** MARBLE exposes specific missing competencies: multimodal perception grounding, visuospatial reasoning, and long-horizon plan consistency - which are not covered by existing benchmarks.
> - **Positioning the direction of frontier multimodal intelligence.** MARBLE is designed precisely to surface the multimodal planning and visuospatial reasoning abilities that are widely acknowledged as essential for future agentic and embodied systems.
>
>
> Finally, it is worth noting that **MARBLE does exhibit meaningful gradations in difficulty**. Simplified subtasks yield substantial spread between models, while the full versions highlight the limits of current perception and reasoning pipelines.
>
> For these reasons, we respectfully argue that the hardness of MARBLE is not a weakness but a deliberate and beneficial design choice. MARBLE fills a critical gap in current multimodal evaluation by demonstrating fundamental limits and providing a durable testbed to spur the development of the next generation of models.

---

### Meta-Review · Area_Chair_a5B1 · 2026-01-07

**Summary:**

This paper proposes a challenging benchmark that focuses on the spatial reasoning and planning capabilities of multimodal LLMs. The benchmark consists of three sets of tasks, requiring models to reason step by step in complex environments, including Portal virtual game surroundings, cube assembly, and path creation and navigation in mazes.

In general, reviewers hold conservative opinions toward the paper. While reviewers agree that the paper presents a series of thoughtful tasks (GE4m, 3WW3, 14V2), a clear data construction pipeline (iioQ, GE4m), and comprehensive evaluations (GE4m, 14V2), they also raise a series of concerns. The major sets of concerns that informed the decisions are as follows:

1. Lack of analysis and potential solutions. Reviewers find that the paper mainly focuses on evaluating model performance and attempting to reveal weaknesses in spatial planning, while providing very little analysis of the underlying failure modes or discussion of potential solutions (GE4m, iioQ). This weakens the overall contributions of the paper.
2. Benchmark components. Reviewer GE4m finds that the paper lacks clear motivation for combining the three tasks into a single spatial reasoning benchmark, as they appear to focus on distinct capabilities. Reviewer 14V2 points out that the environments mostly involve simulated images, which could limit the generalizability of the findings. Similarly, reviewer iioQ raises concerns about the limited distribution of the benchmark. The benchmark could be further improved with a clearer taxonomy and more comprehensive tasks that cover diverse image domains and task domains.
3. Difficulty of the benchmark and the gap between models and humans (14V2, 3WW3). Reviewers find that some tasks may be challenging even for humans, with both models and humans achieving near-zero performance. On the other hand, for easier tasks, the best-performing models show only a small gap or even outperform human performance. As a result, the experimental results do not clearly demonstrate a meaningful gap between multimodal LLMs and humans, and the benchmark may not effectively distinguish model performance.

**Reviewer Concerns:**

The major concerns are partially addressed in the rebuttal. The authors analyze experiments that isolate perception from planning difficulty, study the symbolic inputs, and add confidence intervals for human evaluation. However, some concerns remain.

- Lack of deeper analysis and actionable solutions. While the rebuttal provides additional experiments to support the claim that perception is a key bottleneck, the analysis remains largely descriptive. The implied solution, such as using stronger perception modules, is speculative and not supported by concrete experiments.
- While the rebuttal clarifies the conceptual motivation for combining the three tasks, it does not address concerns about limited domain coverage and distributional diversity, as the benchmark remains focused on simulated environments and narrow task settings.

**Reviewer Scores:**

For GE4m, the concern regarding the lack of actionable suggestions or directions is only partially resolved. While the rebuttal strengthens the analysis of perception as a bottleneck, it does not provide additional experimental validation that translates this analysis into concrete improvement strategies. The score is likely to remain at 4, marginally below the acceptance threshold.

For 3WW3, the rebuttal addresses the reviewer’s major methodological and evaluation concerns, including additional analyses and clarifications. As a result, the score could increase potentially to 6.

For 14V2, the rebuttal clarifies the motivation behind the task design but does not fully address concerns regarding the reliance on simulated environments and the resulting limitations in generalizability. The current experiments do not rule out this limitation, and the score is therefore likely to remain 2 or slightly increase to 4.

For iioQ, the rebuttal does not fully resolve the concern regarding insufficient insight into why MLLMs fail on the benchmark. Although perception is identified as a key bottleneck, this claim is not validated through experiments that incorporate stronger perception modules. The score is therefore likely to remain at 4.

---

### Decision · Program_Chairs · 2026-01-26

Reject